# FindMeIfYouCan: Bringing Open Set Metrics to *near*, *far* and *farther* Out-of-Distribution Object Detection

## Abstract

Recently, out-of-distribution (OOD) detection has gained traction as a key research area in object detection (OD), aiming to identify incorrect predictions often linked to unknown objects. In this paper, we reveal critical flaws in the current OOD-OD evaluation protocol: it fails to account for scenarios where unknown objects are ignored since the current metrics (AUROC and FPR) do not evaluate the ability to find unknown objects. Moreover, the current benchmark violates the assumption of non-overlapping objects with respect to in-distribution (ID) classes. These problems question the validity and relevance of previous evaluations. To address these shortcomings, first, we manually curate and enhance the existing benchmark with new evaluation splits—semantically *near*, *far*, and *farther* relative to ID classes. Then, we integrate established metrics from the open-set object detection (OSOD) community, which, for the first time, offer deeper insights into how well OOD-OD methods detect unknown objects, when they overlook them, and when they misclassify OOD objects as ID—key situations for reliable real-world deployment of object detectors. Our comprehensive evaluation across several OD architectures and OOD-OD methods show that the current metrics do not necessarily reflect the actual localization of unknown objects, for which OSOD metrics are necessary. Furthermore, we observe that semantically and visually similar OOD objects are easier to localize but more likely to be confused with ID objects, whereas *far* and *farther* objects are harder to localize but less prone to misclassification.

## 1 Introduction

In the last decade, the rise of deep learning has introduced prominent breakthroughs and achievements in object detection (OD) (Zou et al., 2023), where models are usually trained under a closed-world assumption: test-time categories are the same as the training ones. However, during deployment in the real world, OD models will encounter Out-of-Distribution (OOD) objects (Nitsch et al., 2021), *i.e.*, object categories different than those observed during training. While facing OOD objects, one of two safety-critical (high-risk) situations can arise: either the unknown objects are incorrectly classified as one of the In-Distribution (ID) classes, or the OOD objects will be ignored (Dhamija et al., 2020).

In response to these safety challenges, researchers have developed two primary approaches: Out-of-Distribution Object Detection (OOD-OD) (Du et al., 2022b) and Open-Set Object Detection (OSOD) (Dhamija et al., 2020). OOD-OD focuses on identifying predictions that do not belong to the ID categories, while OSOD actively attempts to detect the unknown objects themselves. Though both approaches address the fundamental problem of encountering objects from a different semantic space than the training distribution, they employ significantly different methodologies, evaluation metrics, and benchmarks. This methodological divergence has led to isolated research communities and evaluation frameworks that fail to capture the complete picture of model performance when encountering unknown objects.

Currently, the evaluation of OOD-OD relies on a single benchmark, to the best of our knowledge: the VOS-benchmark Du et al. (2022b). The fundamental assumption of this benchmark is that none

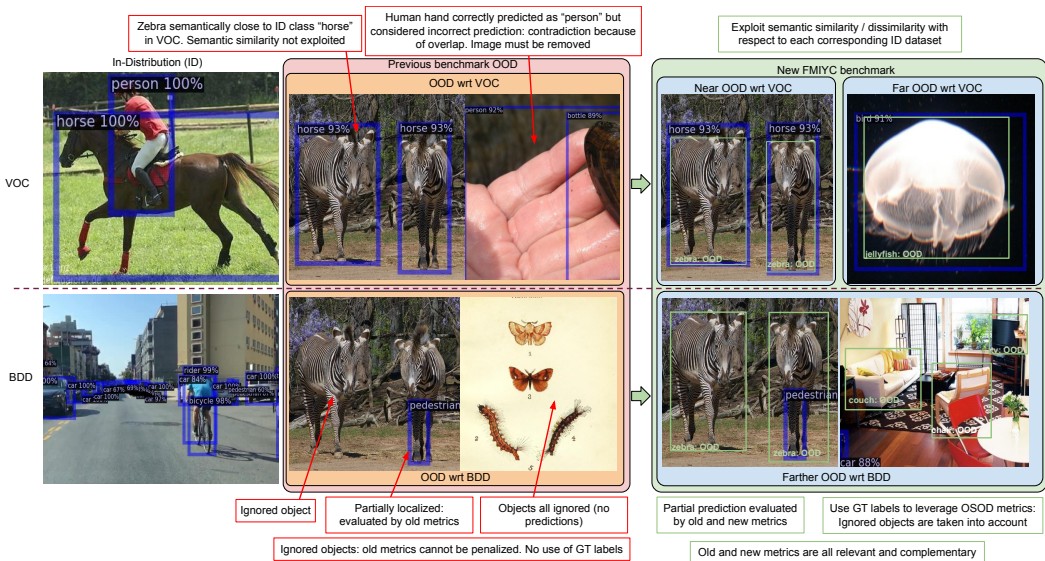

Figure 1: Predictions of Faster-RCNN trained on two ID datasets on samples from each ID and the OOD datasets in blue rectangles. The first row contains predictions of the Faster-RCNN trained on Pascal-VOC. The second row contains the predictions by the model trained on BDD100k. Ground Truth (GT) labels are shown in clear green. The base model predictions are the inputs to OOD scoring functions; without predictions, objects in images will be ignored by OOD scoring functions too. The proposed FMIYC benchmark removes undesirable semantic overlaps and separates semantically *near*, *far*, and *farther* objects with respect to the ID dataset. FMIYC uses ground truth bounding boxes to leverage OSOD metrics that measure when unknown objects are ignored, when they are detected, and when they are confounded with ID objects.

of the images in the OOD datasets include any of the ID classes, implying non-overlapping semantic spaces. Consequently, any prediction made on the OOD datasets by a model trained on the ID classes is inherently incorrect, regardless of the accuracy of object localization. The benchmark employs the area under the ROC curve (AUROC) and the false positive rate at 95% true positive rate (FPR95) as metrics. However, these metrics can be misleading, as they might suggest that a higher AUROC or lower FPR95 indicates better localization of unknown objects, which is not necessarily true. The current benchmark metrics evaluate how well OOD-OD methods identify incorrect predictions, which may potentially correspond to unknown objects. Yet, they fall short of measuring the actual identification of unknown objects. This raises a critical question: *Are AUROC and FPR95 sufficient metrics for assessing the deployment of OOD-OD methods in real-world scenarios?*

In this study, we identify and address fundamental flaws in the existing OOD-OD benchmark and its metrics, while bridging the gap between OOD-OD and OSOD research communities. We demonstrate that the current evaluation violates the fundamental assumption of non-overlap, as the OOD datasets contain ID classes. The benchmark may give the misleading impression of evaluating the identification of unknown objects, fails to penalize ignored unknown objects, and lacks proper assessment of object localization precision–issues that cannot be overlooked for safety-critical applications. To address these challenges, we propose *FindMeIfYouCan* (FMIYC), a comprehensively curated benchmark that: (1) eliminates undesired semantic overlaps between ID and OOD datasets, (2) introduces semantically stratified *near*, *far*, and *farther* OOD splits to evaluate detection robustness across varying levels of semantic similarity, and (3) properly evaluates the actual identification of unknown objects by integrating complementary metrics from the OSOD community, thus providing a robust OOD-OD evaluation framework. By combining strengths from both approaches, our benchmark enables fair comparison across multiple architectures (Faster R-CNN, YOLOv8, RT-, OWLv2) and reveals insights previously obscured in the current standard benchmark. Additionally, we adapt OOD detection methods from image classification and evaluate prominent OOD-OD methods as strong baselines for both OOD-OD and OSOD tasks, establishing a solid foundation for future research that can benefit from both perspectives.

**Contributions.** In summary, the main contributions of this work are:

- We identify and address fundamental flaws in the existing OOD-OD evaluation methodology, demonstrating how the current approach fails to capture a complete picture of the model's performance when encountering unknown objects.

- We propose *FindMeIfYouCan*, a benchmark that removes the existing semantic overlaps and introduces stratified *near*, *far*, and *farther* OOD splits for OOD-OD evaluation across varying levels of semantic similarity.

- We reveal the limitations of legacy AUROC and FPR95 metrics and integrate complementary metrics from the OSOD community for a comprehensive OOD-OD evaluation that captures disregarded objects.

- We assess various methods and architectures for OOD-OD. In particular, post-hoc methods from image classification, and prominent OOD-OD methods. Additionally, we expand the range of evaluated architectures, including the YOLOv8, RT-DETR, and OWLv2 architectures alongside the commonly utilized Faster R-CNN, thereby establishing robust baselines for OOD-OD.

## 2 BACKGROUND & RELATED WORK

### 2.1 OBJECT DETECTION

An object detector is a model $\mathcal{M}$ that takes as input an image $x$ and generates a bounding box $\boldsymbol{b}_i$ and classification score $\boldsymbol{c}_i$ for each $i$-th detected object from a predefined set of categories $\mathbb{C}$ (Girshick et al., 2014). Such models are trained to localize the objects that belong to the ID classes $\mathbb{C}$ and, simultaneously, ignore the rest of the objects and the background (Dhamija et al., 2020). Consequently, the object detector is usually set to function according to a given confidence threshold $t^*$ that corresponds to the one that maximizes the mAP with respect to the ID test dataset. All objects below such threshold $t^*$ are discarded. The model output is the set of tuples $\mathcal{M}(x; t^*) = \{(\boldsymbol{b}_i, \boldsymbol{c}_i)\}$. In the remainder of the paper, the terms "unknown" and "OOD" objects are used interchangeably, and refer to classes that do not belong to $\mathbb{C}$. Two problems can arise during real-world deployment when the model encounters an unknown object: it can be incorrectly detected as one of the ID classes with confidence above the confidence threshold $t^*$, or the unknown object may be ignored. Therefore, two approaches exist in the literature to address these problems: OOD-OD and OSOD.

### 2.2 OOD-OD & OSOD BENCHMARKS

Similar to OOD detection for image classification, OOD-OD is formulated as a binary classification task, that for each detected instance $(\boldsymbol{b}_i, \boldsymbol{c}_i)$ leverages a confidence scoring function $\mathcal{G}$ with its own threshold $\tau$ to calculate a per-object score $\mathcal{G}(\boldsymbol{b}_i, \boldsymbol{c}_i)$ that can distinguish between ID and OOD detections. Du et al. (2022b) introduced a benchmark that has been adopted by subsequent works (Du et al., 2022a; Wilson et al., 2023; Wu & Deng, 2023). This benchmark utilizes BDD100k (Yu et al., 2020) and Pascal-VOC (Everingham et al., 2010) as ID datasets, along with subsets of COCO (Lin et al., 2014) and Open Images (Kuznetsova et al., 2020) as OOD datasets. Trained models on the ID datasets are then set to perform inference on the OOD datasets.

The proposed evaluation method is deemed consistent if it adheres to the critical condition that no ID class appears in any image within the OOD datasets. Consequently, any detection within these OOD datasets is automatically classified as "incorrect", irrespective of whether the prediction corresponds to a ground truth OOD object. Conversely, all predictions on the test ID dataset are considered "correct". By employing this approach, the binary classification metrics AUROC and the FPR95 are utilized to assess the efficacy of the OOD detection method. Specifically, these metrics evaluate how effectively $\mathcal{G}(\boldsymbol{b}_i, \boldsymbol{c}_i)$ assigns different scores to predictions coming from the ID and the OOD datasets (Du et al., 2022b).

On the other hand, OSOD directly adds an *unknown* class to the object detector, along with the ID classes for the training process. It was first formalized by Dhamija et al. (2020), and their goal was to tackle the fact that "unknown objects end up being incorrectly detected as known objects, often with very high confidence". Moreover, the authors propose a benchmark and associated metrics, where the goal is to accurately detect known (ID) and unknown objects simultaneously, as measured by the metrics described in Section 4.2.

The benchmarking setup of OSOD is quite different from that of OOD-OD since, in this setting, the goal is to actively and correctly localize OOD and ID objects at the same time. Also, for OSOD, there is not one commonly accepted benchmark, but many benchmarks have appeared (Ammar et al., 2024; Miller et al., 2018; Han et al., 2022; Dhamija et al., 2020). The common rule is that there is one training dataset with a given set of labeled categories of objects (usually VOC, with 20 categories (Everingham et al., 2010)), and there is one or several subsets of an evaluation dataset that contains the training categories and other labeled classes, semantically different from the ID ones (usually from COCO (Lin et al., 2014)).

# 3  PITFALLS OF THE CURRENT OOD-OD BENCHMARK

**Metrics.**  The current benchmark uses the AUROC and the FPR95 metrics inherited from the image classification task. A misconception that may be conveyed by these metrics is that a higher AUROC or lower FPR95 means better localization of OOD objects, which is not necessarily the case. These metrics measure how well OOD-OD methods identify incorrect predictions, which may or may not correspond to ground-truth unknown objects. Therefore, these metrics do not evaluate the correct localization of OOD objects, and cannot measure when OOD objects are ignored. Figure 2 depicts such issues. For more details on the metrics, see Section C from the Appendix.

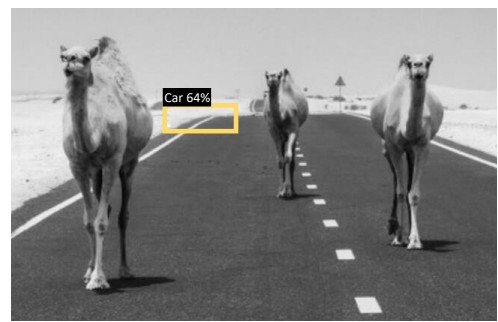

Figure 2: AUROC and FPR95 do not assess whether the relevant unknown objects, such as camels, are overlooked. They only consider incorrect predictions, such as misidentifying a car.

**Semantic overlaps.**  The validity of previously reported results is undermined by the presence of semantic overlaps, as the OOD-OD benchmark fundamentally assumes that no ID objects appear in any OOD dataset. Under this assumption, all model predictions on OOD datasets should be considered incorrect. However, this core assumption is violated, as demonstrated in Figure 1: both labeled and unlabeled instances of people and parts of people are present in the OOD datasets. To maintain benchmark consistency, all OOD images containing ID classes must be removed. For a comprehensive list of overlapping categories in each OOD dataset and further examples, refer to Section A from the Appendix.

**Ignored objects.**  As shown in Figure 1, not all images in each OOD dataset receive at least one prediction. Table 1 reveals that up to 59% of images in one OOD split lack any prediction above the threshold $t^*$. Consequently, the AUROC and FPR95 metrics reported in prior studies, such as Du et al. (2022b); Wilson et al. (2023); Du et al. (2022a); Wu & Deng (2023), are computed using only about 40% of the images in that split. By design, the benchmark's metrics are not penalized for this omission, effectively ignoring a significant portion of images and objects. To address this limitation, we advocate for the adoption of the OSOD metrics introduced in Section 4.2.

Table 1: Percentage of images with no predictions in the current OOD-OD benchmark. OI=OpenImages

| Model | ID: VOC | ID: BDD |
|---|---|---|
| | OI/COCO | OI/COCO |
| F-RCNN | 27.43/35.81 | 59.23/45.27 |
| F-RCNN VOS | 24.08/32.58 | 53.72/40.43 |

**Lack of use of ground truth labels.**  Accurate localization of ground truth (GT) unknown objects is a critical aspect that current benchmarks overlook. A robust evaluation of a system's handling of unknown objects must go beyond simply detecting incorrect predictions. While identifying false positives is important, ignoring unknown objects can be just as risky as misclassifying them (see Figure 2). The OSOD community has established metrics to assess how well methods localize unknowns and to quantify cases where unknowns are either overlooked or confused with in-distribution (ID) objects. To further refine this evaluation, we advocate for the use of GT labels in conjunction with the OSOD metrics outlined in Section 4.2, enabling a more granular and insightful analysis.

# 4 THE FMIYC BENCHMARK

## 4.1 CREATING THE EVALUATION SPLITS

Our newly proposed FMIYC benchmark is built on top of the previous one (Du et al., 2022b), by refining and enriching it in terms of overlap removal, addition of new images, splitting into subsets according to semantic similarity w.r.t. ID datasets, and the addition of open set metrics. All these factors enable fine-grained evaluation of OOD-OD. The first step involved removing overlaps. An automated process first eliminated all labeled instances of overlapping categories. Next, a manual review ensured that no unlabeled ID category instances remained in the datasets.

Then, building on established approaches in OOD detection for image classification–where OOD datasets are divided into semantically and visually *near* and *far* subsets (Zhang et al., 2024; Yang et al., 2023)–we partitioned our OOD datasets w.r.t. Pascal-VOC using class names as the criterion. We matched Pascal-VOC categories (e.g., television, dog, cat, horse, cow, couch) with semantically and visually similar OOD classes (e.g., laptop, fox, bear, jaguar, leopard, cheetah, zebra, bed), assigning these to the *near* subset. All remaining OOD images, lacking a close ID counterpart, were classified as *far*. The splits were validated using WordNet (Miller, 1995) and the Wu-Palmer similarity metric (Wu & Palmer, 1994), with results in Table 9 (Appendix Section B) confirming the stratification. A manual review further ensured that no near-category instances remained in the *far* subset, and vice versa. This process was applied to both COCO and OpenImages, yielding four distinct OOD subsets: COCO-near, COCO-far, OpenImages-near, and OpenImages-far. A complete list and discussion of the *near* OOD categories is available in Appendix Section A.

Table 2: Number of images in each subset of the newly proposed benchmark. CC=COCO, OI=OpenImages

| ID | OOD | No. Images |
|---|---|---|
| VOC | CC Near | 1174 |
| | OI Near | 908 |
| | CC Far | 938 |
| | OI Far | 1179 |
| BDD | CC Farther | 1873 |
| | OI Farther | 1695 |

We selectively incorporated additional images from the original COCO and OpenImages datasets to enrich the newly created *near* and *far* splits. The whole process was documented by recording image IDs in configuration files for each subset, ensuring full reproducibility. Both the code for generating these splits and the resulting datasets will be made publicly available.

For BDD100k as the in-distribution (ID) dataset, only overlapping images were removed, without creating separate *far* or *near* subsets or adding new images. This decision is justified by the findings in Figure 9a and Table 9, which demonstrate that BDD100k is already more distant from its respective OOD datasets than Pascal-VOC. Visual examples illustrating the semantic and visual similarity across all ID and OOD datasets are provided in Appendix Section A. These observations allow us to define three degrees of similarity between ID and OOD datasets: *near* and *far* for OOD datasets relative to Pascal-VOC, and—based on Table 9, Figure 9b, and our

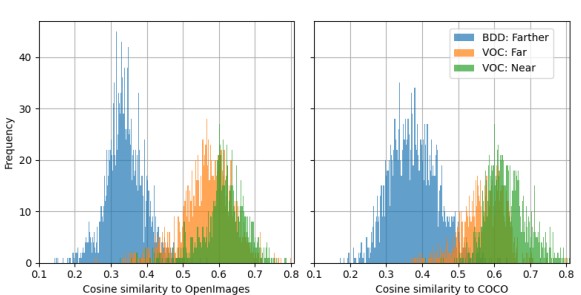

Figure 3: Perceptual and semantic (cosine) similarity (Mayilvahanan et al., 2023) between ID and OOD datasets using CLIP image encoder embeddings.

results—*farther* for OOD datasets relative to BDD100k. The number of images in each subset of the new benchmark is detailed in Table 2. Additionally, we assessed the similarity of each new split with respect to ID datasets in the image space using CLIP vision embeddings, as shown in Figure 3.

## 4.2 PROPOSED METRICS

**OSOD Metrics.** The OSOD community uses as metrics the *absolute open-set error* (AOSE), the *wilderness impact* (WI), the *unknown precision* ($P_U$), *unknown recall* ($R_U$), and the *average precision of the unknowns* $AP_U$ (Gupta et al., 2022; Miller et al., 2018; Maaz et al., 2022). The AOSE reports the absolute number of unknown objects incorrectly classified as one of the ID classes. WI evaluates the proportion of AOSE among all the known detections. Unknown recall $R_U$ is the ratio

of unknown detected objects by the number of unknown ones, and the unknown precision $P_U$ is the ratio of true positive detections divided by all the detections (Ammar et al., 2024). The OSOD metrics are fine-grained in the sense that they assess how well the methods can localize and correctly classify known and unknown objects in images where both types of objects appear.

In addition to the widely used metrics of AUROC and FPR95, we propose using the following OSOD metrics: $AP_U$, $P_U$, and $R_U$. We omit the WI since our benchmark does not allow both ID and OOD classes in the OOD datasets. In addition, we propose a new metric that we call *normalized open set error* (nOSE), which is the AOSE divided by the total number of labeled unknowns. We propose this metric since the absolute number of unknowns depends on the dataset, and therefore, the AOSE is not comparable across datasets, whereas the nOSE is. The nOSE assesses the proportion of unknown objects detected as one of the ID classes. A summary of the overall metrics used in the FMIYC benchmark can be found in Appendix Section C.

## 5 EXPERIMENTS AND RESULTS

### 5.1 OBJECT DETECTION ARCHITECTURES

We used the Faster-RCNN (Girshick et al., 2014) in its *vanilla* and VOS (regularized) versions, YOLOv8 (Jocher et al., 2023; Sohan et al., 2024) and RT-DETR (Zhao et al., 2024). As an extension, we include results from OWLv2 (Minderer et al., 2024), which is a state-of-the-art VLM for object detection. For YOLOv8 and RT-DETR, the models were trained on the same ID datasets (Pascal-VOC and BDD100k). The training details can be found in Appendix Section E. For the Faster-RCNN models, we used the pre-trained checkpoints provided by Du et al. (2022b). For OWLv2, we used the original pretrained model (Minderer et al., 2024). Table 3 shows the architectures mAP for each ID test dataset.

Table 3: mAP across architectures for VOC & BDD ID datasets

| Model | VOC | BDD |
|---|---|---|
| F-RCNN | 48.7 | 31.20 |
| F-RCNN VOS | 48.9 | 31.30 |
| Yolov8 | 54.73 | 32.15 |
| RT-DETR | 70.4 | 33.30 |
| OWLv2 | 73.2 | 30.40 |

### 5.2 OUT-OF-DISTRIBUTION OBJECT DETECTION METHODS

We implemented prominent methods from OOD detection literature on image classification. Specifically, we selected *post-hoc* methods, as they do not require retraining of the base model. Consequently, we adapted the common families of methods from image classification to operate at the object level, as detailed below.

**Output-based post-hoc methods** take the logits, or the softmax activations, as inputs to their scoring functions. Here we can find MSP (Hendrycks & Gimpel, 2016), energy score (Liu et al., 2020), and and GEN (Liu et al., 2023). **Feature-space post-hoc methods** use the previous-to-last activations as the input to the scoring functions. To this category belong kNN (Sun et al., 2022), DDU (Mukhoti et al., 2023) and Mahalanobis (Lee et al., 2018). **Mixed output-feature-space post-hoc methods** rely on the previous-to-last activations and the outputs as the input to the scoring functions. Here we find ViM (Wang et al., 2022), ASH (Djurisic et al., 2022), DICE (Sun & Li, 2022), and ReAct (Sun et al., 2021). **Latent-space post-hoc methods** take inspiration from recent works (Yang et al., 2023; Mukhoti et al., 2023; Arnez et al., 2024) and implement an adapted confidence score, called LaRD, that uses latent activations of a given intermediate or hidden layer.

Adapting *post-hoc* methods for object detection is straightforward, leveraging each architecture's built-in filtering mechanisms. In YOLOv8, however, only MSP, GEN, and energy-based methods are applied, as the network lacks a final fully connected layer or object-specific latent features. In addition to the adapted *post-hoc* OOD detection methods, we evaluated prominent OOD-OD methods such as VOS (Du et al., 2022b), SAFE (Wilson et al., 2023), and SIREN (Du et al., 2022a). The confidence score threshold for each OOD detection method was calculated such that 95% of the ID samples lie above the threshold. Furthermore, as a baseline for OSOD methods in our benchmark, and to enable a fair comparison with OOD-OD methods, we present results for OpenDet CWA (Mallick et al., 2024), a state-of-the-art OSOD method based on Faster-RCNN.

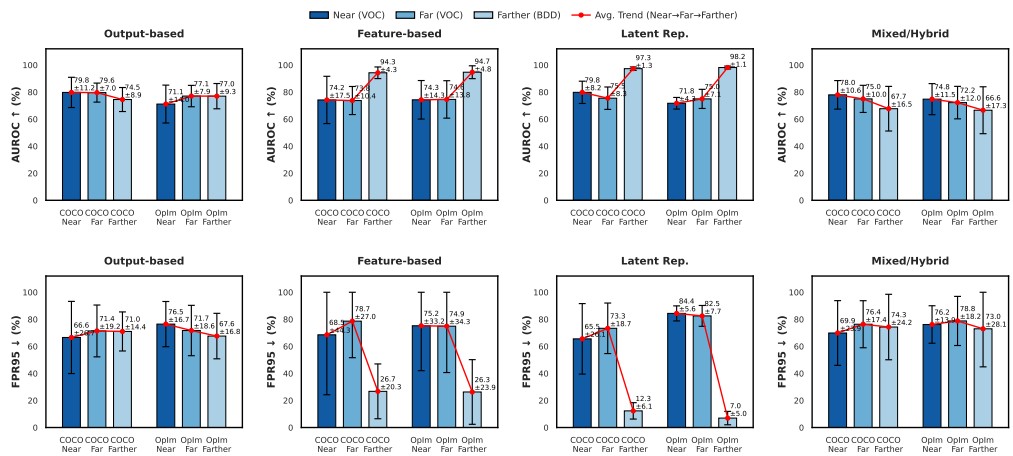

Figure 4: Average OOD-OD performance across baseline families and classic metrics (architectures are averaged). OpIm=OpenImages

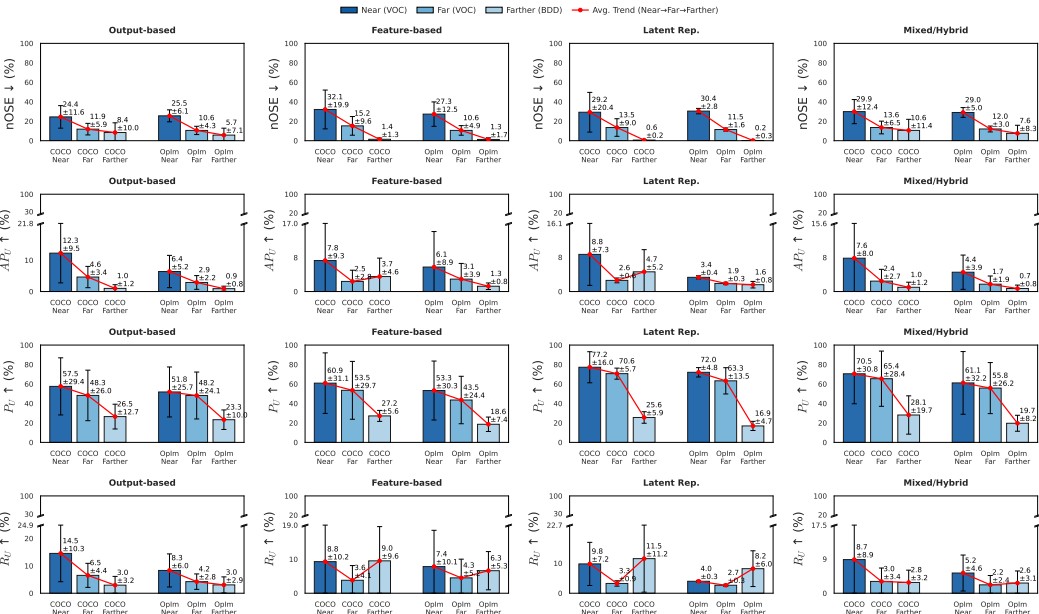

Figure 5: Average OSOD performance comparison across baseline families and metrics (architectures are averaged). OpIm=OpenImages

## 5.3 RESULTS

In Figure 4, we present a summarized plot of the AUROC and FPR95 metrics from the new FMIYC benchmark, averaged across architectures for each family of methods and each OOD dataset. Feature-based methods and those utilizing latent representations tend to identify incorrect predictions more effectively in the *farther* split compared to other splits. Conversely, mixed methods exhibit a decline in performance as semantic distance increases. Overall, there is no distinct trend among baseline families indicating whether incorrect detections are more easily identified for *near*, *far*, or *farther* objects. This observation may be surprising; however, the differences among splits will become more apparent when considering the OSOD metrics discussed subsequently.

Figure 5 illustrates the results for the incorporated OSOD metrics, averaged across architectures for each family of methods and each OOD dataset. For the nOSE, there is a clear decreasing trend across method families when transitioning from *near* to *farther* splits. The *near* datasets exhibit the

Table 4: Results on the COCO datasets for methods using Faster-RCNN (top) and OWLv2 (bottom). **Bold:** best OOD-OD method

| Method | AUROC ↑ | | | $R_U$ ↑ | | | $P_U$ ↑ | | | nOSE ↓ | | |
|---|---|---|---|---|---|---|---|---|---|---|---|---|
| | Near | Far | Farther | Near | Far | Farther | Near | Far | Farther | Near | Far | Farther |
| GEN | 87.43 | 84.48 | 78.82 | 26.12 | 10.96 | 2.99 | 73.80 | 65.17 | 22.89 | **14.29** | **8.69** | 2.04 |
| Energy | 86.47 | 82.31 | 72.44 | 24.84 | 9.95 | 2.99 | **75.88** | **66.33** | 22.89 | 15.95 | 9.80 | 2.03 |
| VOS | **89.98** | **89.13** | 84.79 | 24.62 | 11.26 | **4.72** | 72.10 | 55.61 | **26.70** | 20.49 | 9.65 | **1.76** |
| SAFE | 83.94 | 79.73 | **90.73** | 16.78 | 6.31 | 2.45 | 54.85 | 45.78 | 20.87 | 35.45 | 18.73 | 3.22 |
| SIREN | 89.63 | 88.00 | - | **27.30** | **12.17** | - | 60.52 | 53.67 | - | 19.46 | 9.84 | - |
| OpenDet CWA | - | - | - | 37.85 | 24.59 | 5.39 | 77.69 | 54.72 | 29.19 | 25.19 | 12.57 | 8.30 |
| OWLv2 Energy | 55.02 | 58.79 | 59.45 | 0.0 | 0.0 | 0.0 | 0.0 | 0.0 | 0.0 | 1.18 | 0.15 | 0.01 |
| OWLv2 Mahalanobis | 61.35 | 89.49 | 99.31 | 0.0 | 0.05 | 0.01 | 0.0 | 2.94 | 3.70 | 1.18 | 0.10 | 0.0 |

Table 5: Results on the OpenImages datasets for methods using Faster-RCNN (top) and OWLv2 (bottom). **Bold:** best OOD-OD method

| Method | AUROC ↑ | | | $R_U$ ↑ | | | $P_U$ ↑ | | | nOSE ↓ | | |
|---|---|---|---|---|---|---|---|---|---|---|---|---|
| | Near | Far | Farther | Near | Far | Farther | Near | Far | Farther | Near | Far | Farther |
| GEN | 82.77 | 83.70 | 79.65 | 16.95 | 6.92 | 3.31 | 72.01 | 68.04 | 21.80 | **15.86** | **5.37** | 0.69 |
| Energy | 81.49 | 81.79 | 73.33 | 15.22 | 6.58 | 3.35 | **73.59** | **70.08** | 22.08 | 18.07 | 5.76 | 0.65 |
| VOS | **84.40** | **86.01** | 88.08 | 12.77 | 7.09 | **5.63** | 64.11 | 67.29 | **26.24** | 22.29 | 6.33 | **0.63** |
| SAFE | 85.18 | 83.33 | **95.10** | 14.9 | 4.31 | 3.18 | 55.70 | 55.38 | 17.17 | 26.86 | 9.36 | 1.36 |
| SIREN | 88.61 | 85.22 | - | **20.88** | **6.527** | - | 60.53 | 59.55 | - | 16.34 | 6.15 | - |
| OpenDet CWA | - | - | - | 27.51 | 14.11 | 5.93 | 73.42 | 62.08 | 32.93 | 19.67 | 5.56 | 8.59 |
| OWLv2 Energy | 56.85 | 59.36 | 48.14 | 0.0 | 0.0 | 0.0 | 0.0 | 0.0 | 0.0 | 6.67 | 0.88 | 0.0 |
| OWLv2 Mahalanobis | 70.84 | 87.67 | 99.55 | 0.68 | 0.17 | 0.0 | 23.28 | 20.58 | 0.0 | 5.98 | 0.71 | 0.0 |

highest nOSE, indicating that more objects are mistakenly predicted as one of the in-distribution (ID) classes among the correctly localized objects. Conversely, objects in the *farther* split are less confounded with ID objects. Regarding the $AP_U$, it is generally observed to be low across OOD datasets, with a trend of decreasing further in the *farther* datasets. This suggests that objects that are semantically *near* are localized more accurately. Feature-based methods and those utilizing latent space representations appear to perform better than other methods for the *farther* objects.

The $P_U$ exhibits the highest variability across methods and also the highest values among the OSOD metrics. It is particularly elevated for the near splits. However, drops drastically for the *farther* objects, indicating that in such splits, more OOD predictions do not correspond to ground truth objects, as illustrated in Figure 2. Finally, the $R_U$ is generally quite low across OOD datasets and methods, with a similar trend showing that objects in *far* and *farther* OOD datasets are harder to detect. The metrics reveal that, on average, most unknown objects are ignored (not found), and this challenge is even more pronounced for *far* and *farther* OOD objects. For the *near* splits, $\sim 14\%$ of unknown objects are correctly identified. This figure drops to approximately 3% in the *farther* splits for output-based and mixed methods. However, feature-based and latent representation methods seem to perform slightly better, identifying $\sim 9\%$ of the unknown objects in the *farther* splits. For a comprehensive presentation of the results for each architecture, method, and metric, please refer to Appendix Section F.

It is important to note how unrelated the previous OOD-OD benchmark metrics may seem with respect to the OSOD metrics. The AUROC and FPR95 cannot actually tell much difference between *far* and *near* datasets. This difference becomes clear in light of the OSOD metrics, which show that, contrary to the case of image classification, for object detection, the semantically and visually closer objects are easier to identify and localize. But when the unknown objects are too different from the ID ones, they will most likely be ignored by the methods and architectures evaluated. These insights are impossible to obtain using only the AUROC and FPR95.

Furthermore, Table 4 and Table 5 show summarized results for COCO/OpenImages with the most widely used architecture for OOD-OD, Faster-RCNN, across the two best post-hoc methods (GEN and Energy) according to our results, and including three OOD-OD training methods: VOS (Du et al., 2022b), SAFE (Wilson et al., 2023), and SIREN (Du et al., 2022a). We include one OSOD method based on Faster-RCNN in order to make a fair comparison, OpenDet CWA (Mallick et al., 2024). The tables show no clear winner in all OOD-OD and OSOD metrics. Across training methods, VOS presents the best AUROC performance in terms of near and far splits, and also shows the best $P_U$, $R_U$, and nOSE in the farther split. When comparing OOD-OD methods with OpenDet

CWA, it is possible to observe that it outperforms all other methods in OSOD metrics, which may not come as a surprise since it is specifically an OSOD method. It is worth clarifying that AUROC is not computable for OpenDet CWA (or OSOD methods in general), since OSOD is not a binary classification task, whereas OOD-OD is.

Finally, Table 4 and Table 5 also show the results for OWLv2 using two post-hoc OOD-OD methods. The results for OWLv2 must be understood considering that, on average, about 93% of the images in all OOD subsets do not have a single prediction, constraining the AUROC results to only around 7% of the evaluation images. This, along with the nOSE, indicates that the VLM makes many fewer incorrect predictions than in the case of Faster-RCNN, Yolov8, and RT-DETR. However, AUROC alone can be misleading. A closer look at $R_U$ and $P_U$ shows that OOD methods applied to OWLv2 fail to detect almost any unknown objects. While the model may internally recognize these objects, its output is strictly confined to the queried ID classes. This aligns with recent analysis by Miyai et al. (2024), which argues that VLMs require specialized OOD approaches that account for their prompt-based input and extensive semantic space.

## 6 DISCUSSION

**The value of OSOD metrics.** We suggest caution to practitioners when relying solely on legacy metrics (AUROC and FPR95) and the former evaluation approach, as it does not take into account ignored objects or images without prediction, resulting in fewer 'valid' images for evaluation independently of the architecture for object detection. It is crucial to note that the OSOD metrics are necessary to quantify the effectiveness of OOD-OD methods in detecting actual OOD objects ($AP_U$ and $P_U$) and accounting for instances when OOD objects are overlooked ($R_U$) or misclassified (nOSE). Unlike AUROC and FPR95, the OSOD metrics provide a more nuanced understanding by addressing confounding unknowns for ID objects, the oversight of OOD objects, and the localization of unknowns. The added value of the OSOD metrics is clearer when considering the semantic stratified splits.

*Near, far* and *farther* **splits.** The partition of the benchmark into *near*, *far*, and *farther* proved insightful and meaningful since it details that semantic similarity plays an important role in the detection ability of different methods and architectures. It is especially insightful how the *near* OOD objects are more easily detectable than *far* and *farther* ones in the case of object detection. This is the opposite of the case of image classification, where *near* classes are considered harder than *far* ones. However, the *near* objects are also more easily confounded with ID objects, in agreement with image classification observations. Moreover, the observation that *far* and *farther* objects are more usually ignored, and therefore are hardly localizable, is demonstrated by the OSOD metrics, as only around 5% of the unknown *farther* objects are localized, as opposed to about 20% for some methods in the *near* datasets. Our work paves the way for newer detection approaches customized to specific semantic similarity requirements and provides a stronger foundation for developing OOO-OD and OSOD methods.

## 7 CONCLUSION

In this work, we identified and addressed fundamental flaws in the existing *de facto* out-of-distribution object detection (OOD-OD) evaluation benchmark and its metrics. To address these flaws, we introduced the *FindMeIfYouCan* benchmark, which builds on top of and refines the existing evaluation framework for OOD-OD. In addition, we propose incorporating open-set object detection metrics to comprehensively assess OOD-OD methods on their ability to identify unknown objects. The proposed benchmark approach offers and facilitates a holistic evaluation, measuring the detection of semantically *near*, *far*, and *farther* objects, instances where objects are overlooked, and cases where objects are misclassified as in-distribution (ID) objects. We believe our work lays a solid foundation for a more rigorous and nuanced evaluation of OOD-OD methods towards a more reliable deployment of object detectors in real-world scenarios.

REPRODUCIBILITY STATEMENT.

We include details throughout the paper that can be used to recreate the dataset and to reproduce our results. In particular, Section 4, and Section B from the Appendix. Upon acceptance, we will make publicly available the code used for dataset creation, the dataset created, and benchmark evaluation code, to ensure reproducibility and adoption of the benchmark.

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

**Appendix**

## A    SEMANTIC OVERLAP AND SIMILARITIES IN PREVIOUS BENCHMARK

As stated in Section 3, the main assumption of the current OOD-OD benchmark is that no ID category can be present in the OOD datasets. This is what we call the no-overlap condition. If this condition is met, it is ensured that all predictions done by a model trained on the ID datasets can be considered "incorrect" predictions. The non-overlap condition can mainly be enforced by manual inspection of OOD datasets, due to the existence of unlabeled instances of several objects.

Table 6: Semantic overlap: Number of OOD images containing ID classes

| ID class | No. Images |
| --- | --- |
| Person (or part) | 106 |
| Dining table | 142 |
| Other | 4 |

A close inspection of the dataset showed that, in fact, the core assumption of no overlap is not met, since there are labeled and unlabeled instances of ID categories in the OOD datasets. The amount of images in the OOD datasets that contain ID categories is shown in Table 6.

Figure 6: Examples of images in the OOD datasets that contain humans or parts of humans. There exists a semantic overlap between ID and OOD datasets. The images must be removed for the benchmark to have consistency.

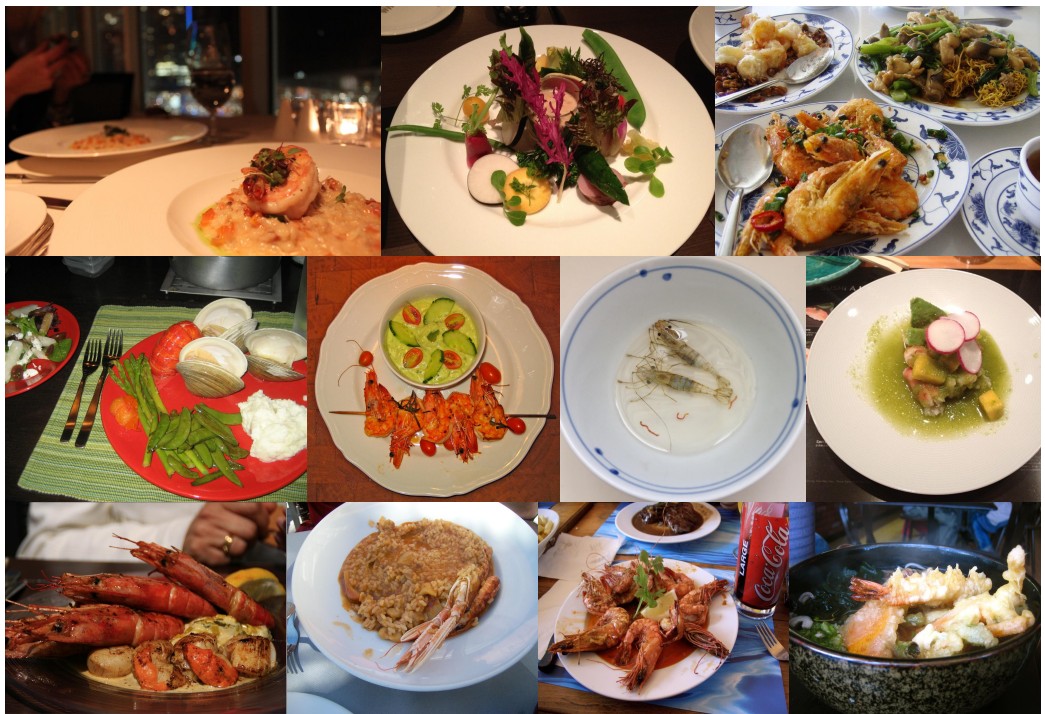

Figure 7: Examples of images in the OOD datasets that contain dining tables. Some of these contain also humans. There exists a semantic overlap between ID and OOD datasets. The images must be removed for the benchmark to have consistency.

Some examples of images in the OOD datasets that contain humans or parts of humans are shown in Figure 6. Similarly, examples of images containing "dining tables" in the OOD datasets w.r.t. VOC are shown in Figure 7. Table 7 shows the overlapping categories in each OOD dataset.

Table 7: Overlapping categories in each OOD dataset w.r.t. VOC

| ID: VOC | COCO | OpenImages |
|---|---|---|
| Person | Person | Person, human face, human arm, woman, human head, human hand, human hair, human nose, human ear, human mouth, human nose, human eye, human beard, body part |
| Dining table | Spoon, fork, pizza, sandwich, cake, hot dog, wine glass, spoon | Salad, plate, broccoli, tableware, fork, baked goods, spoon |
| Boat | - | Boat |
| Potted plant | - | Houseplant, flowerpot |
| Cat | - | Cat |

All images containing overlapping classes with the ID ones must be removed for the benchmark to comply with the non-overlap condition. Table 7 presents the detailed list of OOD categories that overlap with the corresponding ID category in each OOD dataset with respect to VOC categories. For BDD100k as ID, only the images containing instances of people or parts of people were removed.

Furthermore, we present a list of OOD categories and their corresponding ID category that are considered semantically or visually *near* w.r.t. VOC in Table 8. All the other categories in the OOD datasets that are not in the *near* list are considered *far* categories when VOC is the ID dataset. It is important to note, as explained in Section 4, that the images were manually checked to ensure the correct assignment into each new split, or removal. Figure 8 show examples of OOD images that contain *near* categories w.r.t. VOC as ID dataset, along with the prediction from Faster-RCNN trained on VOC.

Table 8: Semantically and visually *near* categories in each OOD dataset w.r.t. VOC

| VOC category | COCO | OpenImages |
|---|---|---|
| Horse | Zebra | - |
| Cat | - | Jaguar, leopard, cheetah |
| Chair | Bench | - |
| Person | - | Clothing |
| Dining table | Spoon, fork, carrot, orange, apple, cup, bowl | Zucchini, food, knife |
| Television | Laptop | Tablet computer, laptop |
| Couch | Bed | - |
| Dog | Bear | Fox |
| Potted plant | Vase | - |
| Various | - | Raccoon, harbor seal, hedgehog, otter, sea lion |

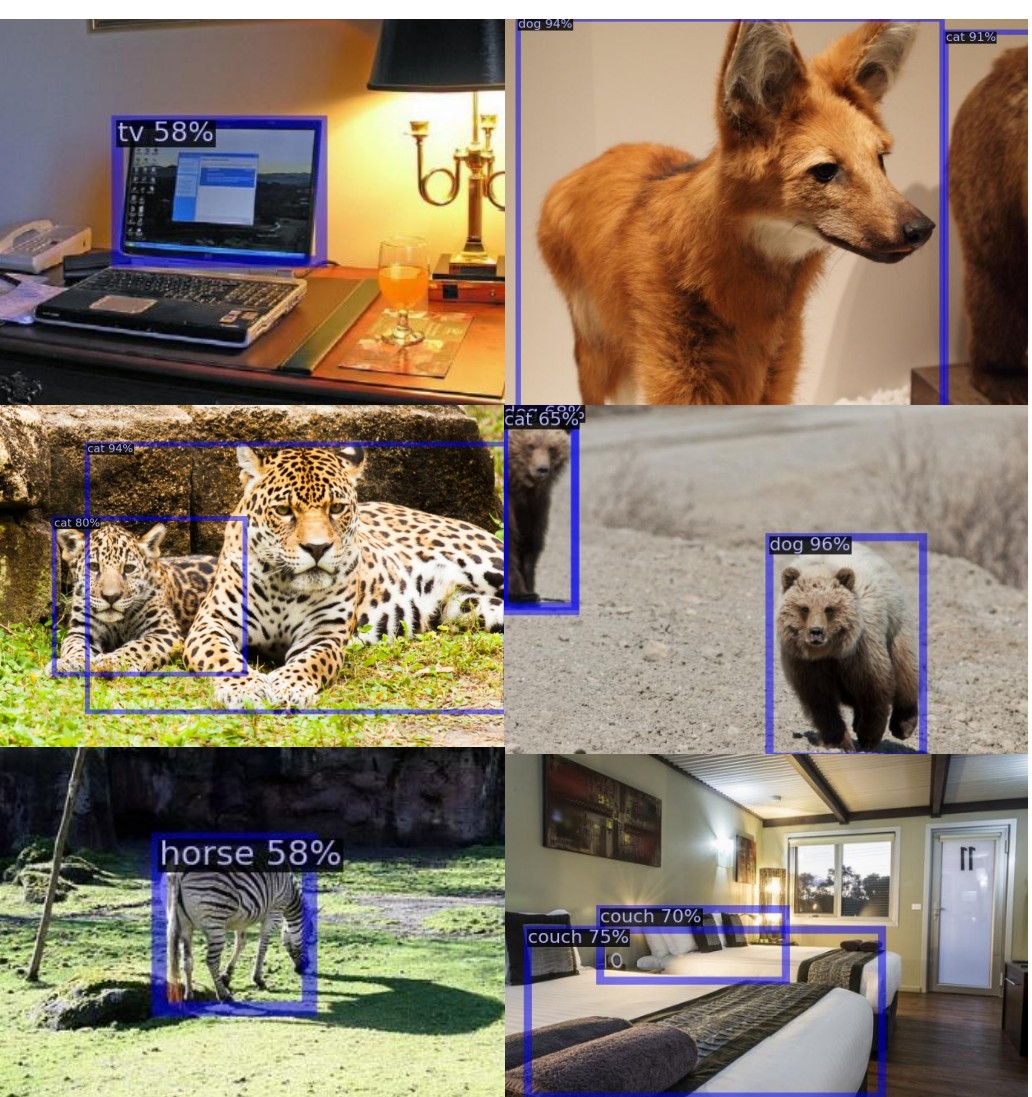

Figure 8: Examples of images in the OOD datasets that contain categories classified as *near* w.r.t. Pascal-VOC as ID dataset. The predictions are made by the Faster RCNN model trained on Pascal-VOC.

# B    DETAILS ON THE CONSTRUCTION OF THE FMIYC BENCHMARK

Here we provide more details into how the new benchmark was created, in addition to what is already presented in Section 4. Following the observations made in Section A with respect to the semantic overlaps existing in the current OOD-OD benchmark (Du et al., 2022b), the first step was to remove the images where semantic overlap exists with the ID categories.

The second step consisted of splitting into *near* and *far* subsets with respect to Pascal-VOC using class names as the criterion. The images containing semantically and visually similar categories from Table 8 were put into the *near* split. The rest were put into the *far* split. The splits were validated using WordNet (Miller, 1995) and the Wu-Palmer similarity metric (Wu & Palmer, 1994). For each class name in the ID and OOD datasets, the WordNet embedding was obtained. Then, we calculated the highest Wu-Palmer similarity of each OOD class name w.r.t. those of the ID class names. The results in Table 9 show the average WuP similarity for each proposed split, and confirm the stratification. The images were manually inspected to ensure no unlabeled instances of ID categories

Table 9: Wu-Palmer average similarity scores for the proposed splits. CC=COCO, OI=OpenImages

| ID | OOD dataset | WuP similarity |
|---|---|---|
| VOC | CC Near | 0.706 ± 0.225 |
| | OI Near | 0.642 ± 0.204 |
| | CC Far | 0.683 ± 0.177 |
| | OI Far | 0.604 ± 0.193 |
| BDD | CC Farther | 0.619 ± 0.158 |
| | OI Farther | 0.508 ± 0.175 |

were present, in which case the image was removed from the benchmark. The manual inspection also ensured the correct assignment of images to each split.

Next, new images were added to each split. Candidate images from the training sets of COCO and OpenImages were first selected for manual inspection. The candidate images didn't have labeled ID categories, and needed to contain labeled instances of either the *near* or the *far* categories. Candidate images for each split were then manually inspected to ensure also that no ID category was present, and the correct assignment to each split.

For BDD100k as ID, the only modification done to the existing OOD datasets was the removal of images with people, because of overlap with the ID category "pedestrian".

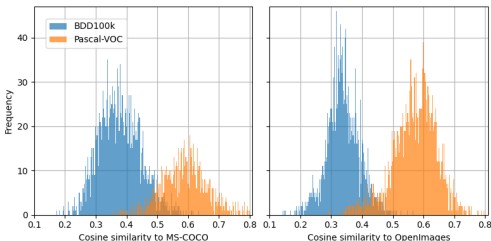 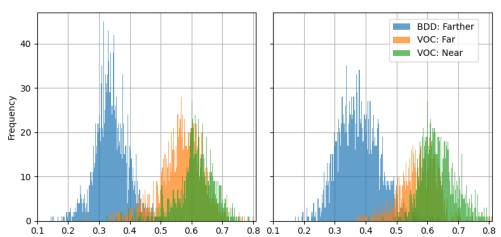

(a) Current benchmark: VOC is semantically and visually more similar to OOD datasets than BDD.

(b) The FMIYC benchmark distinction of *near*, *far* and *farther* splits can be appreciated

Figure 9: Perceptual and semantic (cosine) similarity (Mayilvahanan et al., 2023) between ID and OOD datasets using CLIP image encoder embeddings.

Later, the semantic and visual similarity was assessed using CLIP (Radford et al., 2021) embedding space. The embeddings for both ID datasets, and for OOD samples in each split were extracted. Then, following the procedure in Mayilvahanan et al. (2023), we calculated the cosine similarity between ID and their respective OOD datasets. The obtained results before and after creating the splits can be seen in Figure 9. It can be observed that three groups are present. This allowed us to propose the distinction into *near*, *far* and *farther* datasets. *Near* and *far*, are splits that are OOD w.r.t. VOC. Farther are the subsets w.r.t. BDD100k. Each of these subsets exists for COCO and Open-Images, which means that in total, there are six subsets of OOD datasets: COCO-near, COCO-far, OpenImages-near, OpenImages-far w.r.t. VOC; along with COCO-farther and OpenImages-farther w.r.t. BDD100k. The amount of images in each subset is shown in Table 2. In total, there are 7767 images across all splits.

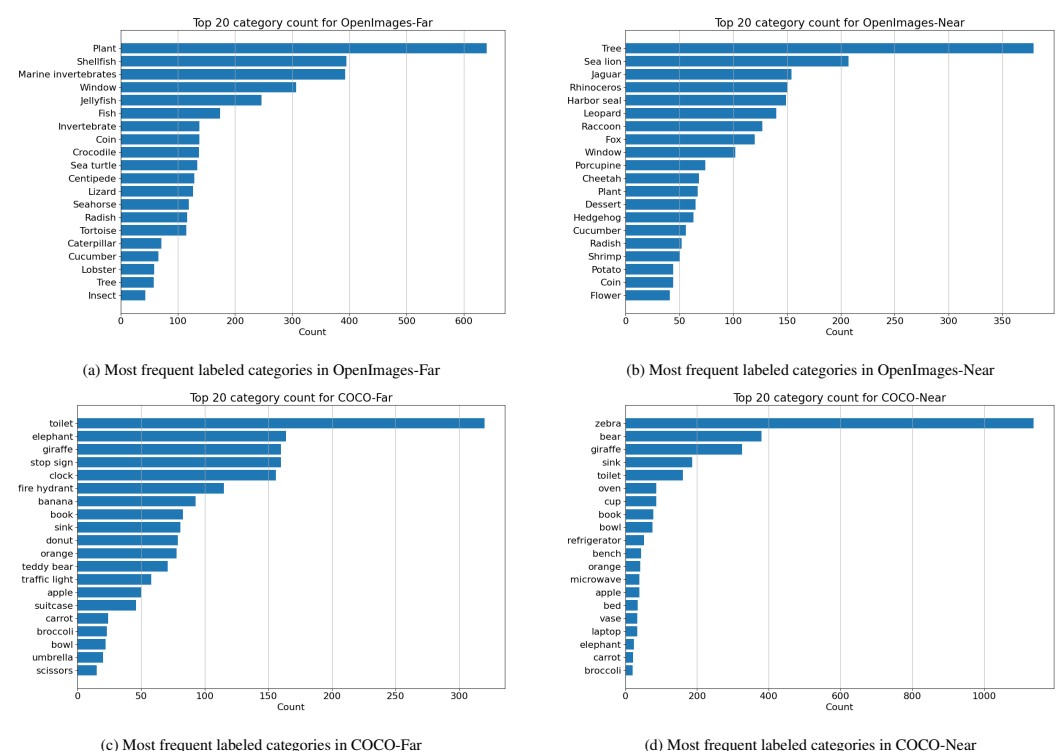

Figure 10: Top 20 category count for OOD datasets w.r.t. Pascal-VOC

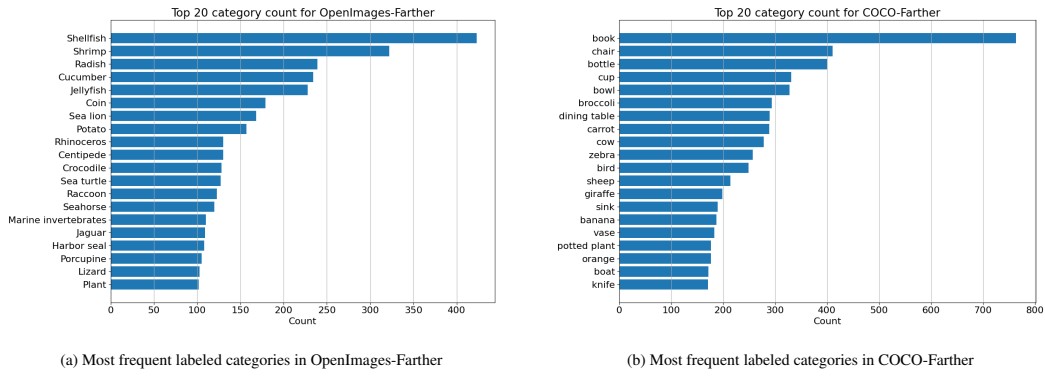

Figure 11: Top 20 category count for OOD datasets w.r.t. BDD100k

Finally, Figure 10 and Figure 11 show the top-20 category count for the images in each split of the new benchmark.

## C  DETAILS ON THE METRICS USED

This section provides more details about the previous and the newly incorporated metrics.

**Previous OOD-OD metrics**   AUROC and FPR metrics come from binary classification problems. The receiver-operating-characteristic (ROC) curve evaluates the performance of a classifier at varying threshold values. It consists of the plot of the true positive rate (TPR) against the false positive rate (FPR) at each threshold setting. TPR and FPR are defined as follows:

$$\text{FPR} = \frac{FP}{FP + TN} \tag{1}$$

$$\text{TPR} = \frac{TP}{TP + FN} \tag{2}$$

where $FP$ is the number of false positives, $TP$ is the number of true positives, $TN$ is the number of true negatives, and $FN$ is the number of false negatives.

The AUROC is the area under the ROC curve. Since both TPR and FPR are bounded to the interval $[0, 1]$, the AUROC is bounded to the same interval. A perfect classifier would have an AUROC of 1, whereas a random classifier would have an AUROC of 0.5. The value of 0 would mean that the classifier is a perfect misclassifier (predicts negatives as positives and vice-versa). The FPR95 is the false positive rate at 95% true positive rate. The lower the FPR95, the fewer false positives the classifier predicts (Lasko et al., 2005).

For the previous OOD-OD benchmark, the main limitation of these two metrics lies in the fact that they have no relation with ground truth (GT) bounding boxes, and rely exclusively on the compliance with the non-overlap assumption, as described in Section 2.2 and Section A. Therefore, AUROC and FPR95 are unable to measure the actual localization of OOD objects. For an illustration of this, see Figure 12.

Moreover, a non-negligible amount of images does not have a single prediction at all, as can be seen in Table 1. AUROC and FPR95 cannot measure that the main objects in Figure 2, Figure 12 and Figure 13 are ignored. They can only take into account the incorrect predictions as in Figure 12. Even if the unknown objects are correctly localized, AUROC and FPR95 are not measuring this since they are unrelated to the GT bounding boxes. For these reasons, we raise the critical question: *are AUROC and FPR95 sufficient metrics to assess the deployment of OOD-OD methods in safety-critical real-world scenarios?*

**OSOD metrics**   The newly proposed metrics for the benchmark exist in the Open Set for object detection (OSOD) community. The metrics were already introduced in Section 4.2. here we give a more detailed definition for each one of them. It is important to note that all of the metrics were calculated using an intersection over union (IoU) threshold of 0.5. This means that one detection is considered as a true positive ($TP_U$) if the unknown is classified correctly (as unknown or OOD), and its predicted bounding box has an IoU$\geq 0.5$ with a ground truth (GT) unknown object.

Also, for this case it is important to distinguish two types of false negatives: dismissed or ignored ones, denoted $FN^D$, and misclassified ones, denoted $FN^M$. One prediction is considered as $FN^D$ if no predicted bounding box has IoU$\geq 0.5$ with the GT label. A detection is considered $FN^M$ if a bounding box has IoU$\geq 0.5$ with a GT unknown but the predicted class is one of the ID categories. The total false negatives for the unknowns are then:

$$FN_U = FN_U^D + FN_U^M \tag{3}$$

The precision of the unknowns $P_U$ is defined in a similar way as the binary classification metric:

$$P_U = \frac{TP_U}{TP_U + FP_U} \tag{4}$$

where all quantities refer to unknowns: $TP_U$ are the true positive predictions, and $FP_U$ are the false positive predictions. Also, let us note that $TP_U + FP_U$ are the total number of predictions

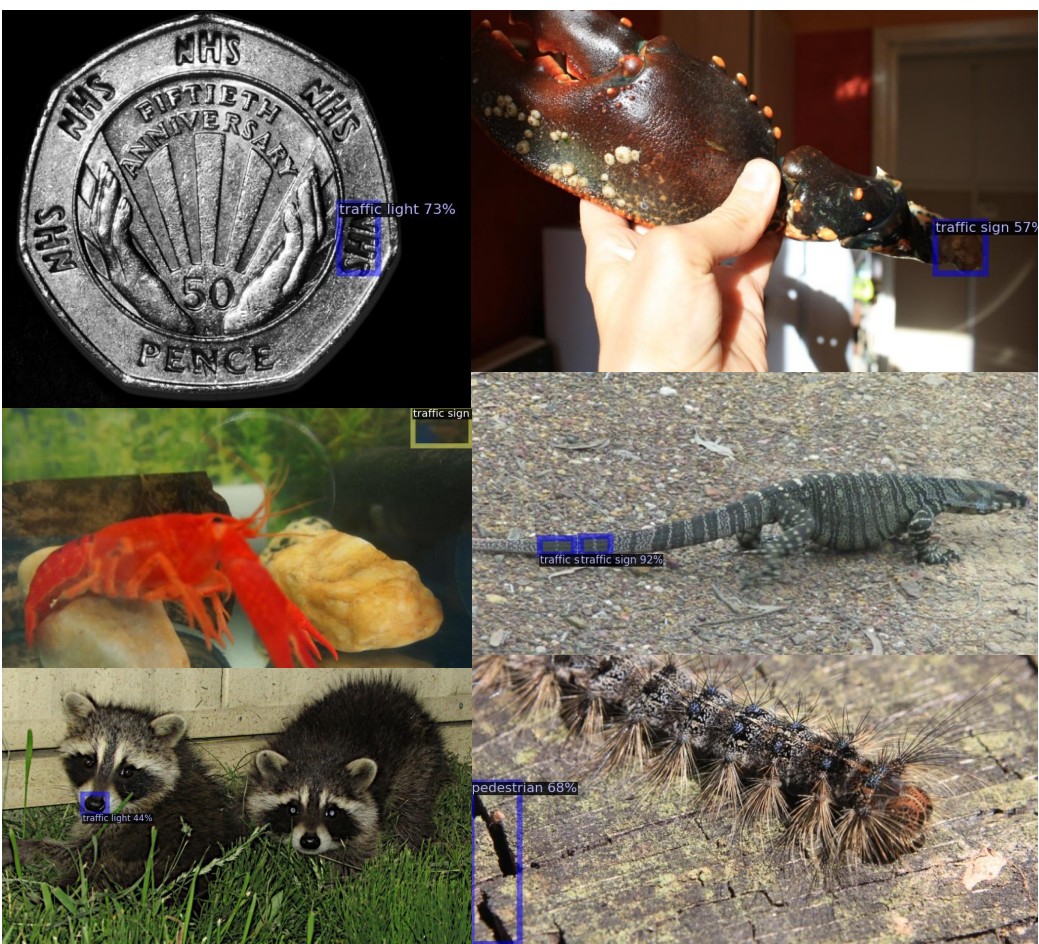

Figure 12: Incorrect predictions of Faster-RCNN trained on BDD100k on images from the OOD datasets in the current benchmark. AUROC and FPR95 cannot measure that the main OOD objects are ignored. They can only take into account the incorrect predictions. OSOD metrics can quantify the dismissal of unknown objects

for the unknown class. Therefore, what $P_U$ is measuring is the ratio of true positives divided by all unknown predictions. In other words, $P_U$ tells the proportion of predictions for unknowns that were actually ground-truth unknowns (Powers, 2011).

The recall of the unknowns $R_U$ is defined as:

$$R_U = \frac{TP_U}{TP_U + FN_U} \tag{5}$$

where $FN_U$ are the false negatives. Let us note that $TP_U + FN_U$ are the total number of ground-truth unknowns. In other words, $R_U$ tells us the proportion of ground-truth unknowns that were found by the detector.

For the average precision of the unkowns $AP_U$, it is defined as the area under the precision-recall curve:

$$AP = \int_0^1 p(r)dr \tag{6}$$

which is usually calculated by the interpolation of rectangles of the sampled values:

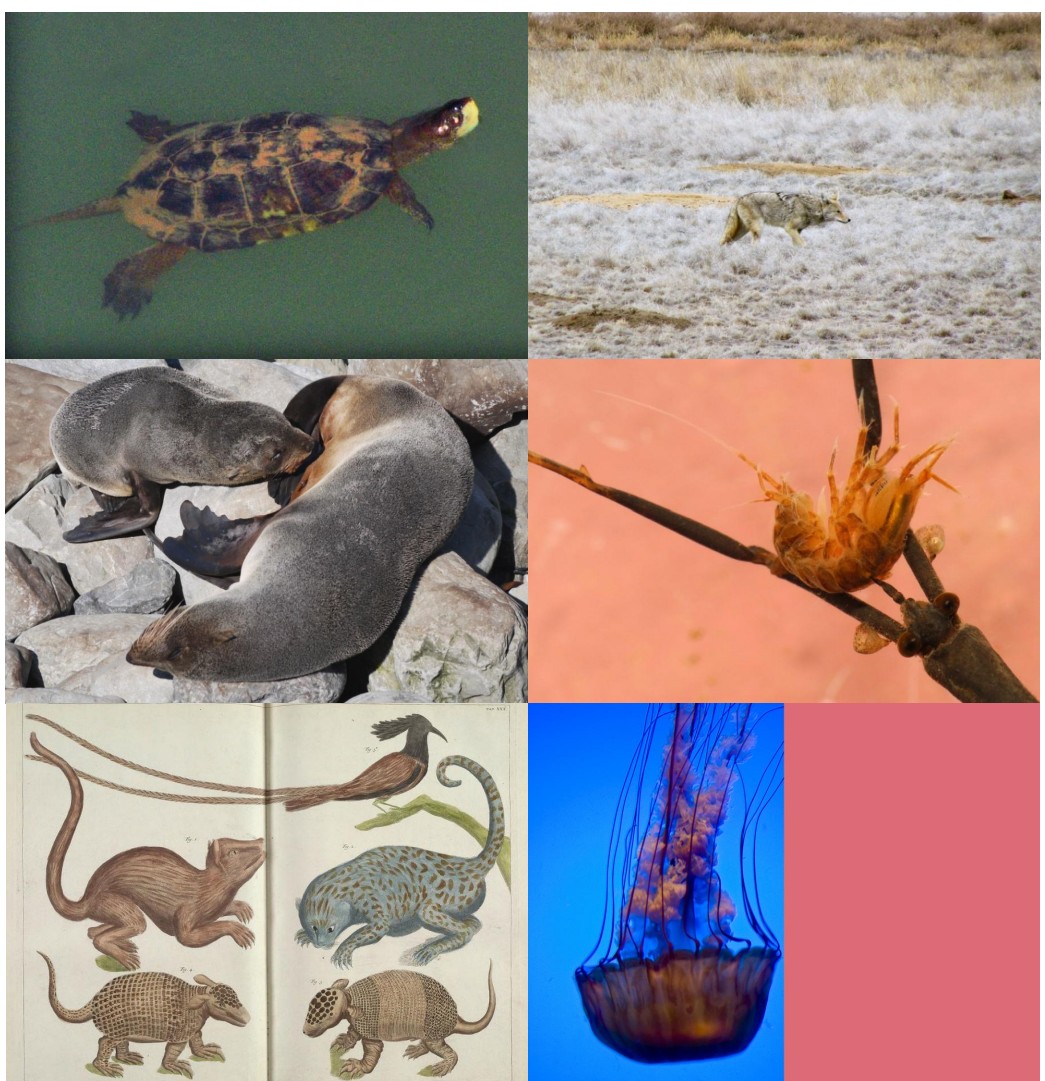

Figure 13: Absense of predictions of Faster-RCNN trained on BDD100k on images from the OOD datasets in the current benchmark. AUROC and FPR95 cannot measure that all OOD objects in these images are ignored. Dismissing OOD objects is not measurable using the current metrics. OSOD metrics can quantify the dismissal of unknown objects

$$AP = \sum_{m}^{M} (r_{n+1} - r_n) p_{in}(r_{n+1}), \tag{7}$$

$$p_{in}(r_{n+1}) = \max_{\tilde{r} \geq r_{n+1}} p(\tilde{r}) \tag{8}$$

where $p_{in}$ represents the interpolated precision at each detection point, which is obtained by taking the maximum precision whose recall value is greater or equal than $(r_{n+1})$ (Padilla et al., 2020).

Next, usually OSOD works report the absolute open set error (AOSE), that is defined as the total number of unknown objects that are predicted as one of the ID classes (which would correspond to $FN_U^M$). Since the absolute number of these is not comparable across datasets (because each dataset has a different number of unknown objects), we propose using a metric that we call normalized open set error (nOSE) that is defined as:

$$\text{nOSE} = \frac{FN_U^M}{TP_U + FN_U} \tag{9}$$

where indeed $TP_U + FN_U$ is once more the total number of ground-truth unknown objects. The nOSE is comparable across datasets, and estimates the proportion of OOD objects that are confounded with ID objects.

A summary of the purpose, limitations, and advantages of the used metrics can be found in Table 10.

Table 10: Overall metrics summary

| Metric | Purpose | Limitations | Advantages |
|---|---|---|---|
| AUROC, FPR95 | Measures the ability of a scoring function to detect incorrect predictions | Cannot take into account ignored objects | Does not depend on GT labels, can detect incorrect predictions that do not overlap with labeled objects |
| Precision | Measures the percent of correct predictions over the total of predictions | Need good GT labels. Cannot measure unlabeled unknowns. | Measure localization of GT objects |
| Recall | Measures the percent of found objects divided by the total number of labeled objects | | |
| nOSE | Measures the percent of unknown objects confounded with an ID object | | |

# D   DETAILS ON EVALUATED OOD DETECTION METHODS

We present further details on the OOD detection methods used in the paper. All of the methods come from the Image classification literature (Yang et al., 2024), except for VOS (Du et al., 2022b).

## D.1   PRELIMINARIES.

Using the notation from Section 2.1, let us recall that a trained object detector $\mathcal{M}$ takes as input an image $x$, along with a confidence threshold $t^*$, and for each $i$-th detected object outputs a bounding box $\boldsymbol{b}_i \in \mathbb{R}^4$ and a vector of logits $\boldsymbol{c}_i \in \mathbb{R}^{|\mathbb{C}|}$, with dimension equal to the number of ID classes $\mathbb{C}$. The model output is the set:

$$\mathcal{M}(x; t^*) = \{(\boldsymbol{b}_i, \boldsymbol{c}_i)\}_{i=1}^D \tag{10}$$

where $D$ is the number of detections in each image. Each tuple $(\boldsymbol{b}_i, \boldsymbol{c}_i)$ corresponds to one detected object. Note that $D = 0$ is possible, and in such a case the output is empty. Furthermore, the so-called softmax activation is given by:

$$\sigma(c_j) = \frac{e^{c_j}}{\sum_m^{|\mathbb{C}|} e^{c_m}} \tag{11}$$

which transforms the logits vector into a vector of probabilities for each ID class, such that $\sum_j^{|\mathbb{C}|} \sigma(c_j) = 1$. In this notation, the index $j$ denotes the class index, and the index $i$ denotes the object index. An alternative output is then given by the vector of probabilities after softmax: $\mathcal{M}(x; t^*) = \{(\boldsymbol{b}_i, \boldsymbol{p}_i)\}_{i=1}^D$, where $p_j = \sigma(c_j)$. The predicted probability of each detected object is the maximum after softmax, let it be denoted by $\hat{p}_i = \max_j p_{ij}$. In any case, to have $D > 0$, there must be at least one prediction such that $\hat{p}_i \geq t^*$.

**The OOD detection problem.**   Is formulated as a binary classification task leveraging a (confidence) scoring function $\mathcal{G}$ for each detected instance $(\boldsymbol{b}_i, \boldsymbol{c}_i)$, so that:

$$\mathcal{G}(x, \boldsymbol{b}_i, \boldsymbol{c}_i) = g_i \in \mathbb{R} \tag{12}$$

The scoring function aims to distinguish between ID and OOD objects, using a thresholding function $\boldsymbol{\Omega}$ with threshold $\tau$ as presented in eq. (13).

$$\boldsymbol{\Omega}\Big(g_i, \tau\Big) = \begin{cases} 1 & ID & \text{if } g_i \geq \tau \\ 0 & OOD & \text{if } g_i < \tau \end{cases} \tag{13}$$

For the OOD-OD problem, only those detected objects above the threshold $t^*$ are considered. Therefore, if no object is detected in a given image, there is no input for the scoring function $\mathcal{G}$ for such an image. In a general sense, each of the OOD detection methods is a realization of the scoring functions $\mathcal{G}$. Figure 14 presents a depiction of the workflow of OOD-OD scoring functions.

It is important to avoid possible confusion and it can be useful to reiterate here that $t^*$ and $\tau$ are two different thresholds. The object detection model $\mathcal{M}$ uses a confidence threshold $t^* \in \mathbb{R}^{[0,1]}$ that is usually the one that maximizes the mAP in the ID test set. This threshold filters the output of the model so that all detected objects satisfy $\hat{p}_i \geq t^*$. On the other hand, the OOD scoring functions $\mathcal{G}$ use each one its own threshold $\tau \in \mathbb{R}$, which corresponds to the one that makes that 95% of the $g_i$ of detected ID objects are above the threshold.

## D.2   EVALUATED METHODS

For the adaptation of each method from image classification to object detection, in each case, the score is calculated per each detected object above the threshold $t^*$. Therefore, there can be zero or several detections per image. Each of the equations in the following section has been adapted to match our notation, and all of them explain the adaptation done to work at the object level.

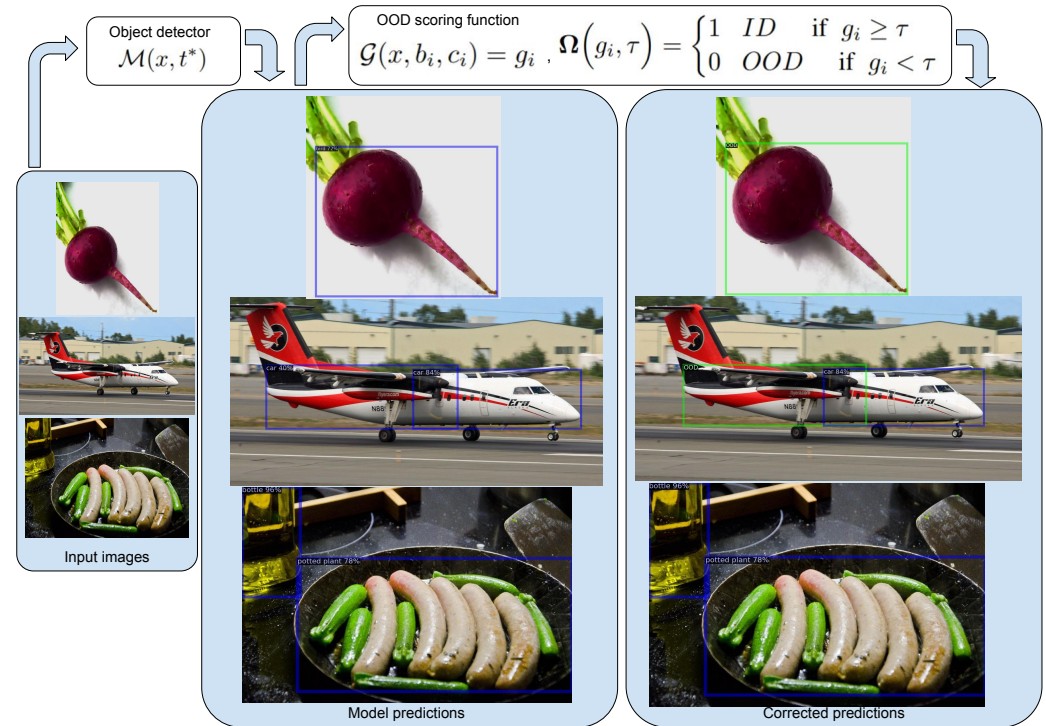

Figure 14: General workflow of OOD-OD scoring functions. The outputs of the base model $\mathcal{M}$ are the inputs to scoring functions $\mathcal{G}$. If the object detector ignores a given object, scoring functions will ignore it, too. The model predictions not marked as OOD, remain with the predicted class.

### D.2.1 OUTPUT-BASED METHODS

Output based methods take either the $c_i$ or the $p_i$ as input to the scoring functions. This family of methods is applicable to all of the architectures tested: Faster-RCNN, Yolov8 and RT-DETR.

**Maximum softmax probability (MSP).** This is perhaps the most classical baseline in OOD detection for image classification (Hendrycks & Gimpel, 2016). It consists of directly choosing the maximum softmax value:

$$\max_j p_j = \max_j \frac{e^{c_j}}{\sum_m^{|\mathcal{C}|} e^{c_m}} \tag{14}$$

where $e$ is the Euler number.

**Energy score.** Proposed by Liu et al. (2020), it calculates the energy score using the activation logits $c_i$ as:

$$E(\boldsymbol{c}_i; T) = -T \log \sum_j^{|\mathbb{C}|} e^{c_j/T} \tag{15}$$

where $T$ is the temperature (usually set to $T = 1$).

**Generalized entropy score (GEN).** Presented by Liu et al. (2023), the authors propose using the family of generalized entropies:

$$G_\lambda(\boldsymbol{p}_i) = \sum_j p_j^\lambda (1 - p_j)^\lambda \tag{16}$$

when $\lambda = 1/2$:

$$G_{1/2}(\boldsymbol{p}_i) = \sum_j \sqrt{p_j(1 - p_j)} \tag{17}$$

### D.2.2 FEATURE-BASED METHODS

If the model $\mathcal{M}$ has $L$ total layers, and its last layer $L$ is a linear one (also called fully connected), then the activations of the $L-1$ (penultimate) layer are considered the extracted features $\boldsymbol{z}_{L-1} \in \mathbb{R}^d$, where $d$ is the dimension of the feature. Then, for a given input image $x$, and a detection $(\boldsymbol{b}_i, \boldsymbol{c}_i)$, the features of each detected object are defined as:

$$\boldsymbol{z}_{L-1}^i = \mathcal{M}_{L-1}(x; t^*) \tag{18}$$

where $\mathcal{M}_l$ denotes the latent activation of $\mathcal{M}$ at layer $l$. To simplify notation, let us denote the per-object feature $\boldsymbol{z}_{L-1}^i$ by $\boldsymbol{z}_i$. In all cases, $\boldsymbol{z}_i^*$ denotes the features of a detected object $(\boldsymbol{b}_i^*, \boldsymbol{c}_i^*)$ from a test image $x^*$. Feature-based methods considered here need a training phase, and for this phase they take as input the $\boldsymbol{z}_i$ of the training set. At test time, their input is the $\boldsymbol{z}_i^*$ of test samples.

This family of methods is not applicable to Yolov8, since this architecture has no final linear layer: it is fully convolutional. Therefore, it is not possible to associate a set of features to a specific detected object. This family of methods can be used with Faster-RCNN and RT-DETR.

**k-Nearest neighbors (kNN).** Introduced by Sun et al. (2022), first normalizes the feature for each detected object: $\mathbf{z}_i = \boldsymbol{z}_i/\|\boldsymbol{z}_i\|_2$, where $\|\cdot\|_2$ denotes the L2 norm. Then, the normalized embeddings of the training data are stored: $\overline{\mathbb{Z}}_N = (\mathbf{z_1}, ..., \mathbf{z}_N)$, where $N$ are the number of objects detected in the training set.

During testing, the normalized features $\mathbf{z}_i^*$ are derived, and the euclidean distances $\|\mathbf{z}_i^* - \mathbf{z}_j\|_2$ are calculated with respect to the train embeddings $\mathbf{z}_j \in \mathbb{Z}_N$. Afterward, the embeddings are reordered according to the increasing distance $\|\mathbf{z}_i^* - \mathbf{z}_j\|_2$. The reordered embedding sequence is $\overline{\mathbb{Z}}_N' = (\mathbf{z_{(1)}}, \mathbf{z_{(2)}}, ..., \mathbf{z}_{(N)})$. The scoring function is defined as:

$$r_k(\mathbf{z}_i^*) = \|\mathbf{z}_i^* - \mathbf{z}_{(k)}\|_2 \tag{19}$$

which corresponds to the distance to the k-th nearest neighbor in the normalized feature space (Sun et al., 2022).

**Mahalanobis distance.** Proposed by Lee et al. (2018), the Mahalanobis score calculates the distance to the centroids of a class-conditional Gaussian distribution. The predicted class per detected object is denoted $y_c^i$ and corresponds to the index of the $\max$ value of either the $c_i$ or the $p_i$. Then the empirical class mean and covariance matrix of training samples are estimated:

$$\hat{\mu}_c = \frac{1}{N_c} \sum_{j:y_c} \boldsymbol{z}_j, \quad \hat{\Sigma} = \frac{1}{N} \sum_c^{\mathcal{C}} \sum_{j:y_c} (\boldsymbol{z}_j - \hat{\mu}_c)(\boldsymbol{z}_j - \hat{\mu}_c)^\top \tag{20}$$

where $N_c$ denotes the total number of objects of class $y_c$ detected in the training set, $N$ is the total number of detected objects in the training set in all classes, and $j$ are de indexes of detected objects of class $y_c$. Then the Mahalanobis confidence score is defined as the Mahalanobis distance between the features $z_i^*$, and the closest class-conditional Gaussian distribution:

$$M(\boldsymbol{z}_i^*) = \max_c -(\boldsymbol{z}_i^* - \hat{\mu}_c)\hat{\Sigma}^{-1}(\boldsymbol{z}_i^* - \hat{\mu}_c)^\top \tag{21}$$

which corresponds to the log of the probability of the test sample (Lee et al., 2018).

**Deep deterministic uncertainty (DDU).**    A work by Mukhoti et al. (2023), fits a Gaussian mixture model (GMM) on the feature space, then computes the density under the GMM. Similar to Equation (20), the mean per class $\hat{\mu}_c$ and the covariance matrix $\hat{\Sigma}$ are computed for the features $z_i$ of each detected object $(b_i^*, c_i^*)$. Then the weights of the GMM are computed as:

$$\pi_c = \frac{1}{N} \sum y_c \tag{22}$$

which denotes the proportion of detected objects for each class $y_c$ over the total $N$ detected objects in the training dataset. During inference time, the density under the GMM is computed for the features $\boldsymbol{z}_i^*$ of a detected object $(\boldsymbol{b}_i^*, \boldsymbol{c}_i^*)$ from a test image $x^*$:

$$q(\boldsymbol{z}_i^*) = \sum_{y_c} q(\boldsymbol{z}_i^*|y_c)\pi_c, \quad \text{where} \quad q(\boldsymbol{z}_i^*|y_c) \sim \mathcal{N}(\mu_c; \sigma_{y_c}) \tag{23}$$

### D.2.3    OUTPUT-FEATURE (MIXED) BASED METHODS

This family of methods takes both the outputs (either the $c_i$ or the $p_i$) and the features $z_i$ for each detected object $(b_i, c_i)$ as inputs to the scoring functions. This family of methods was not applicable to Yolov8 for the same reasons as for the previous family of methods.

**Activation shaping (ASH).**    Showcased by Djurisic et al. (2022), involves a reshaping of the feature $z_i$, and subsequent use of the energy score from Equation (15). The reshaping is done by first calculating a threshold $t$ that corresponds to the $p$-th percentile of the entire set of the detected objects representations of the training set:

$$\mathbb{Z}_N = (z_1, ..., z_N) \tag{24}$$

Afterward, we calculate $s_1 = \sum_j z_j$. Then all values below $t$ are set to 0 to obtain a pruned version of the features $\mathbb{Z}_N^p = (z_1^p, ..., z_N^p)$. Using the $\mathbb{Z}_N^p$, we calculate $s_2 = \sum_j z_j^p$. Finally, all non-zero values in $\mathbb{Z}_N^p$ are multiplied with $\exp(s_1/s_2)$, to obtain the pruned and reshaped features:

$$\begin{aligned} \mathbb{Z}_N^r &= \mathbb{Z}_N^p \exp(s_1/s_2) \\ &= (z_1^p \exp(s_1/s_2), ..., z_N^p \exp(s_1/s_2)) \\ &= (z_1^r, ..., z_N^r) \end{aligned} \tag{25}$$

Finally, the pruned and reshaped features are passed through the final fully connected layer $L$ to obtain the logit activations $c_i$, which are passed to the energy score calculation as in Equation (15). The authors found that the method works best when using a pruning percentile of about 90% (Djurisic et al., 2022).

**Directed sparsification (DICE).**    Introduced by Sun & Li (2022), the authors consider the weight matrix of the final fully connected layer $\mathbf{W} \in \mathbb{R}^{d \times |\mathcal{C}|}$, where $d$ is the dimension of the feature $z_i$, and $|\mathcal{C}|$ is the number of ID categories. This matrix is then subject to sparsification, to preserve the most important weights in it. The contribution is measured by a matrix $\mathbf{V} \in \mathbb{R}^{d \times |\mathcal{C}|}$, where each column $\mathbf{v}_c \in \mathbb{R}^d$ is given by:

$$\mathbf{v}_c = \mathbb{E}_{z_j \in \mathbb{Z}_N}[\mathbf{w}_c \odot z_j] \tag{26}$$

where $\odot$ represents the element-wise multiplication, $\mathbf{v}_c$ indicates the weight vector for class $y_c$, and $\mathbb{Z}_N$ is as defined in Equation (24). Then the top-$k$ weights are selected from the largest values of $\mathbf{V}$, to obtain a sparsified matrix $\mathbf{W}'$. This matrix is now used as the final layer weights instead of the $\mathbf{W}$. Finally, the obtained $c_i$ are passed to the energy scoring function from Equation (15) (Sun & Li, 2022).

**Rectified activations (ReAct).**    Proposed by Sun et al. (2021), it performs a clipping operation on the features $z_i$, and the calculation of the energy score. The rectification (or clipping) is performed as:

$$\bar{\boldsymbol{z}}_i = \min(\boldsymbol{z}_i, t) \tag{27}$$

where each element of $\boldsymbol{z}_i$ is truncated to be at most equal to the threshold $t$. This threshold is calculated so that a given percentile of the activations is less than the threshold. For instance, at percentile $p = 90$, 90% of ID train activations are below the threshold $t$. The authors found that a percentile of 90 works best. Then, the $\bar{\boldsymbol{z}}_i$ are passed as inputs to the final layer to obtain the outputs $\boldsymbol{c}_i$, which are then used to calculate the energy score as in Equation (15) (Sun et al., 2021).

**Virtual logit matching (ViM).**    A method inspired by a thorough geometrical analysis of the space of the matrix $\mathbf{Z}$, whose rows are the $\boldsymbol{z}_i$ for all detected objects in the training set. Let $\mathbf{X}$ denote a centered version of $\mathbf{Z}$, obtained by offsetting the $\boldsymbol{z}_i$ by a vector $\mathbf{o} = -(\mathbf{W}^\top)^+\mathbf{b}$, where $(\cdot)^+$ denotes the Moore-Penrose inverse, $\mathbf{W}$ is the final layer weight matrix and $\mathbf{b}$ is the final layer bias. The eigendecomposition of the matrix $\mathbf{X}^\top\mathbf{X}$ is:

$$\mathbf{X}^\top\mathbf{X} = \mathbf{Q}\boldsymbol{\Lambda}\mathbf{Q}^{-1} \tag{28}$$

where eigenvalues $\boldsymbol{\Lambda}$ are ordered decreasingly. The first $D$ columns of $\mathbf{Q}$ are called the $D$-dimensional principal subspace $P$. The residual subspace $P^\perp$ is spanned by the remaining $D+1$ to the last columns of $\mathbf{Q}$, and is represented by the matrix $\mathbf{R} \in \mathbb{R}^{N \times (N-D)}$, where $N$ is the number of detected objects in the train set. Then $\boldsymbol{z}_i^{P^\perp}$ denotes the projection of $\boldsymbol{z}_i$ onto $\mathbf{R}$: $\boldsymbol{z}_i^{P^\perp} = \mathbf{R}\mathbf{R}^\top z_i$. The virtual logit $c_0$ is calculated as:

$$c_0 = \alpha\|\boldsymbol{z}_i^{P^\perp}\| = \alpha\sqrt{\boldsymbol{z}_i^\top\mathbf{R}\mathbf{R}^\top\boldsymbol{z}_i} \tag{29}$$

which corresponds to the norm of the residual $z_i^{P^\perp}$ rescaled by a constant $\alpha$. This constant is calculated as:

$$\alpha = \frac{\sum_j^K \max_{m=1,\ldots,|\mathcal{C}|}\{c_m^j\}}{\sum_{j=1}^K \|\boldsymbol{z}_i^{P^\perp}\|} \tag{30}$$

where $\boldsymbol{z}_1, \boldsymbol{z}_2, \ldots, \boldsymbol{z}_K$ are uniformly sampled $K$ training examples, and $c_m^j$ is the $m$-th logit of $c_j$. This constant scales the virtual logit to the average maximum of the original logits. Finally, the ViM score is calculated as:

$$\text{ViM}(z_i) = \alpha\|z_i^{P^\perp}\| - \ln\sum_{j=1}^{|\mathbb{C}|} e^{c_j} \tag{31}$$

which, in summary, is the virtual logit minus the energy score of the rest of the logits. For the hyperparameter $D$, the authors recommend using $D = 1000$ if the dimension of the feature $d > 1000$, or use $D = 512$ otherwise (Wang et al., 2022).

### D.2.4  LATENT SPACE METHODS

In this family we find methods that take as input other latent activations inside the network. We took inspiration from Arnez et al. (2024); Wilson et al. (2023) and built a method based on the latent space convolutional activations. In our case, we used directly the latent activations without doing Monte Carlo dropout sampling of entropy estimation as in Arnez et al. (2024), nor using a surrogate model or the generation of adversarial examples as in Wilson et al. (2023).

**Latent representation density (LaRD).** We start by considering a convolutional feature map $z_{i,l} \in \mathbb{R}^{N_c \times W \times H}$, where $N_c$ is the number of channels, $W$ is the width and $H$ is the height of the latent activation, extracted at layer $l$. Then it is possible to use the predicted bounding boxes $b_i$ and the feature maps as inputs for the ROIAlign (RA) algorithm (He et al., 2017), which can extract the corresponding portion of the feature maps per each predicted object:

$$o_{i,l} = \mathrm{RA}(z_{i,l}, b_i), \text{where } o_{i,l} \in \mathbb{R}^{N_c \times R \times R} \tag{32}$$

Where $R$ is the parameter that fixes the size of the output of the RA algorithm, that outputs crops of the feature map $z_{i,l}$ with a given fixed-sized for all objects, independently of their aspect ratio or actual size in the image. Then an average per channel is taken to reduce the dimensionality of these representations:

$$\bar{o}_{i,l} = \frac{1}{HW} \sum_{h=1}^{H} \sum_{w=1}^{W} o_{i,l}(c, h, w), \text{ where } \bar{o}_{i,l} \in \mathbb{R}^{N_c} \tag{33}$$

The set $O_l = \{\bar{o}_{i,l}, y_i\}_{d=1}^{D}$ consists of all the averaged latent representations at layer $l$ of each object found by the object detector in one image, along with the predicted class $y_i$. Then, we also want to build a density estimator, by making a forward pass through the training set to obtain the set of all the ID objects latent representations: $\mathcal{O}_{train,l} = \{O_l\}_{x=1}^{N_t}$, where $N_t$ is the size of the training set. Afterward, we use the methodology as in the Mahalanobis distance baseline to obtain a scoring function for each of the detected objects. We used a hyperparameter of $R = 9$ for all experiments. For Faster-RCNN, the chosen latent layer was the RPN intermediate convolutional layer as in Arnez et al. (2024); for Yolov8, it was the final layer of the backbone, after evaluation of each layer. For RT-DETR the chosen hidden layer was the first encoder module, similarly, after evaluation of each layer.

# E  DETAILS ON THE TRAINING OF ARCHITECTURES

This section provides details on the training of Yolov8 (Sohan et al., 2024) and RTDETR (Zhao et al., 2024). Both architectures were trained on a single GPU Nvidia A100 40G. The achieved mAP by both models in each ID dataset is found in Table 3.

## E.1  YOLOV8

We trained the nano version of Yolov8 for both ID datasets (BDD100k and Pascal-VOC). We used the same hyperparameters for both models. Most of them corresponded to the default hyperparameters. They were trained for 100 epochs, using the AdamW optimizer with momentum of 0.937 and weight decay of $5 \times 10^{-4}$. The learning rate was $10^{-3}$, and was controlled by a cosine scheduler. The batch size was 16, and we used the copy-paste augmentation, on top of the mosaic, translate, scale, erase, and flip-lr default augmentations. For the training, we used the Ultralytics library (Jocher et al., 2023).

## E.2  REAL-TIME DETR

We fine-tuned a version of RT-DETR that was pre-trained on COCO for both ID datasets (BDD100k and Pascal-VOC). The pretrained version can be found in Huggingface: RT-DETR. Both versions used early stopping with a patience of 16 epochs. The hyperparameters for both models can be found in Table 11.

Table 11: Hyperparameters for training RT-DETR whith ID datasets BDD100k and Pascal-VOC

| Parameter | ID: BDD | ID: VOC |
|---|---|---|
| Batch size | 8 | 8 |
| Inference threshold | 0.25 | 0.25 |
| Learning rate backbone | $4 \times 10^{-6}$ | $2 \times 10^{-6}$ |
| Max epochs | 60 | 60 |
| Num queries | 100 | 100 |
| Random seed | 40 | 40 |
| Learning rate | $4 \times 10^{-5}$ | $2 \times 10^{-5}$ |

# F  DETAILED RESULTS PER METHOD AND ARCHITECTURE

This section presents detailed results for each architecture and method, covering all metrics. First, we present a table showing the number of images without predictions in the proposed benchmark. Then, the results for previous metrics are presented. Afterwards, the results for the new metrics are detailed. Finally, a study of the correlations among previous and new metrics is presented.

Table 12: Percentage of images with no predictions in the proposed OOD-OD benchmark. OI=OpenImages.

|  | Near OI/COCO | Far OI/COCO | Farther OI/COCO |
|---|---|---|---|
| Faster RCNN | 18.28/20.36 | 49.19/55.01 | 59.88/45.22 |
| Faster RCNN VOS | 15.64/19.34 | 44.27/51.49 | 54.28/40.42 |
| Yolov8 | 14.98/18.48 | 30.2/42.32 | 70.15/55.79 |
| RT-DETR | 18.06/49.66 | 38.85/81.66 | 14.16/8.06 |
| OWLv2 | 77.75/94.12 | 94.57/95.31 | 99.65/98.83 |

The previous table shows that the object detector models ignore more images when moving to the *farther* split. Interestingly, the VLM model OWLv2 is the one that ignores the most of the images. This indicates that this model mistakes OOD objects less frequently for ID ones. The metrics presented in Section 5.3 should be interpreted in consideration of this table, as AUROC cannot reflect the amount of data used to build it, which needs to be reported.

For instance, the results from OWL in the further split indicate an AUROC of about 99%. However, this metric is built using only 1% of the images, which corresponds to approximately 20 images. The results of this table illustrate once more the need to quantify how often OOD-OD methods ignore OOD objects, as is one of the core contributions of our paper.

## F.1  DETAILED RESULTS ON THE PREVIOUS OOD-OD METRICS

Table 13: OOD detection performance for FasterRCNN (Vanilla) on various OOD splits (ID: PascalVOC). Metrics are AUC↑ (%) and FPR95↓ (%). LaRD represents best of (Mahalanobis PCA, KNN PCA, GMM PCA). Best result per metric column is in **bold**. [B]Indicates the primary scoring method of the VOS (Virtual Outlier Synthesis).

| | FasterRCNN (Vanilla) — PascalVOC | | | | | | | |
|---|---|---|---|---|---|---|---|---|
| | COCO-Near (OOD) | | COCO-Far (OOD) | | OpImg-Near (OOD) | | OpImg-Far (OOD) | |
| Method | AUC↑ | FPR95↓ | AUC↑ | FPR95↓ | AUC↑ | FPR95↓ | AUC↑ | FPR95↓ |
| ViM | 75.7 | 85.5 | 77.8 | 87.4 | 73.0 | 87.1 | 74.9 | 91.6 |
| Mahalanobis | 59.8 | 95.9 | 64.9 | 95.5 | 59.7 | 94.6 | 60.3 | 95.9 |
| MSP | 73.8 | 88.3 | 77.3 | 88.0 | 70.5 | 90.4 | 75.4 | 87.9 |
| Energy | 86.5 | 45.5 | 82.3 | 56.2 | 81.5 | 57.9 | 81.8 | 52.6 |
| ASH | 82.9 | 49.9 | 74.5 | 66.6 | 78.7 | 59.9 | 74.8 | 60.8 |
| DICE | 82.7 | 62.0 | 78.2 | 76.7 | 79.1 | 67.3 | 76.7 | 71.6 |
| ReAct | 85.1 | 58.1 | 75.2 | 82.5 | 83.1 | 66.0 | 73.4 | 83.0 |
| GEN | 87.4 | 44.8 | 84.5 | 55.0 | 82.8 | **56.2** | 83.7 | 52.1 |
| DICE+ReAct | 66.3 | 89.8 | 56.0 | 94.8 | 71.4 | 88.9 | 48.3 | 99.0 |
| DDU | 64.0 | 97.6 | 68.3 | 97.0 | 70.4 | 97.2 | 66.3 | 98.3 |
| VOS[B](Energy) | **90.0** | **44.6** | **89.1** | **44.9** | **84.4** | 60.0 | **86.0** | **49.1** |
| LaRD | 73.8 | 81.7 | 68.6 | 88.0 | 70.0 | 88.4 | 70.0 | 89.2 |

Table 14: OOD detection performance for FasterRCNN enhanced with VOS (Virtual Outlier Synthesis) on various OOD splits (ID: PascalVOC). Metrics are AUC↑ (%) and FPR95↓ (%). LaRD represents best of (Mahalanobis PCA, KNN PCA, GMM PCA). Best result per metric column is in **bold**. [B]Indicates the primary scoring method of the VOS (Virtual Outlier Synthesis).

| | FasterRCNN (VOS) — PascalVOC | | | | | | | |
|---|---|---|---|---|---|---|---|---|
| | COCO-Near (OOD) | | COCO-Far (OOD) | | OpImg-Near (OOD) | | OpImg-Far (OOD) | |
| Method | AUC↑ | FPR95↓ | AUC↑ | FPR95↓ | AUC↑ | FPR95↓ | AUC↑ | FPR95↓ |
| ViM | 77.4 | 87.7 | 80.3 | 85.9 | 73.4 | 89.8 | 77.2 | 92.2 |
| Mahalanobis | 60.9 | 95.9 | 65.5 | 94.9 | 60.3 | 95.5 | 64.8 | 95.5 |
| MSP | 69.1 | 91.5 | 75.1 | 89.2 | 65.6 | 91.1 | 72.6 | 88.2 |
| ASH | **90.2** | 44.1 | 87.4 | 51.4 | 84.8 | 59.8 | 82.5 | 56.2 |
| DICE | 88.0 | 56.5 | 88.3 | 53.4 | 82.7 | 67.8 | 80.8 | 59.0 |
| ReAct | 87.1 | 57.1 | 79.9 | 72.2 | **85.6** | 64.5 | 77.1 | 76.3 |
| GEN | 89.7 | **42.9** | **89.3** | 45.7 | 85.3 | **58.2** | **86.0** | 50.7 |
| DICE+ReAct | 74.9 | 84.8 | 67.3 | 88.1 | 74.8 | 88.5 | 58.6 | 98.9 |
| DDU | 67.5 | 99.2 | 70.0 | 96.9 | 72.5 | 99.3 | 72.7 | 98.3 |
| VOS[B](Energy) | 90.0 | 44.6 | 89.1 | **44.9** | 84.4 | 60.0 | 86.0 | **49.1** |
| LaRD | 75.1 | 77.5 | 68.1 | 87.8 | 67.8 | 87.2 | 67.8 | 89.2 |

Table 15: OOD detection performance for FasterRCNN variants on Farther OOD splits (ID: BDD). LaRD represents best of (Mahalanobis PCA, KNN PCA, GMM PCA). Higher AUC is better (↑), lower FPR95 is better (↓). Best result per metric column is in **bold**. [B]For the FasterRCNN (VOS) architecture, this indicates the primary scoring method of the VOS (Virtual Outlier Synthesis).

| | FasterRCNN (Vanilla) — ID: BDD | | | | FasterRCNN (VOS) — ID: BDD | | | |
|---|---|---|---|---|---|---|---|---|
| | COCO-Farther (OOD) | | OpImg-Farther (OOD) | | COCO-Farther (OOD) | | OpImg-Farther (OOD) | |
| Method | AUC↑ (%) | FPR95↓ (%) | AUC↑ (%) | FPR95↓ (%) | AUC↑ (%) | FPR95↓ (%) | AUC↑ (%) | FPR95↓ (%) |
| ViM | 91.4 | 39.3 | 91.6 | 39.3 | 92.9 | 32.3 | 93.1 | 31.5 |
| Mahalanobis | 89.5 | 48.8 | 89.0 | 51.5 | 91.1 | 43.3 | 90.6 | 46.7 |
| MSP | 80.0 | 77.7 | 81.2 | 76.8 | 79.1 | 79.4 | 80.0 | 76.6 |
| Energy | 72.4 | 64.4 | 73.3 | 60.3 | — | — | — | — |
| ASH | 48.9 | 81.0 | 49.0 | 77.3 | 67.6 | 70.6 | 71.7 | 61.4 |
| DICE | 68.3 | 69.2 | 69.3 | 65.0 | 77.7 | 57.9 | 71.6 | 49.0 |
| ReAct | 65.7 | 95.1 | 58.8 | 97.4 | 79.6 | 71.2 | 77.0 | 76.4 |
| GEN | 78.8 | 62.7 | 79.6 | 58.9 | 86.6 | 52.7 | 89.5 | 47.8 |
| DICE+ReAct | 57.9 | 97.7 | 48.5 | 98.9 | 66.8 | 90.5 | 59.4 | 95.5 |
| DDU | 90.8 | 41.6 | 91.5 | 42.6 | 92.2 | 37.2 | 92.9 | 40.1 |
| VOS[B](Energy) | 84.8 | 49.1 | 88.1 | 38.5 | 84.8 | 49.1 | 88.1 | 38.5 |
| LaRD | **96.6** | **15.8** | **97.7** | **8.6** | **96.6** | **15.8** | **97.4** | **10.9** |

Table 16: OOD detection performance for YOLOv8 (ID: PascalVOC). LaRD represents results from available PCA methods (KNN PCA 32 only in provided data). Higher AUC is better (↑), lower FPR95 is better (↓). Best result per metric column is in **bold**.

| | YOLOv8 — PascalVOC | | | | | | | |
|---|---|---|---|---|---|---|---|---|
| | COCO-Near (OOD) | | COCO-Far (OOD) | | OpImg-Near (OOD) | | OpImg-Far (OOD) | |
| Method | AUC↑ (%) | FPR95↓ (%) | AUC↑ (%) | FPR95↓ (%) | AUC↑ (%) | FPR95↓ (%) | AUC↑ (%) | FPR95↓ (%) |
| MSP | **85.2** | **64.0** | 81.4 | 73.7 | **85.1** | **67.4** | 82.0 | 74.4 |
| Energy | 57.0 | 95.2 | 66.1 | 91.3 | 51.6 | 96.1 | 65.6 | 92.4 |
| GEN | 81.3 | 65.0 | 79.5 | **67.2** | 81.0 | 68.9 | **82.3** | **59.1** |
| LaRD | 78.6 | 76.4 | **82.0** | 68.8 | 71.4 | 85.7 | 80.9 | 75.7 |

Table 17: OOD detection performance on Farther OOD splits (ID: BDD). LaRD for RT-DETR represents best of (Mahalanobis PCA, KNN PCA, GMM PCA). Higher AUC is better (↑), lower FPR95 is better (↓). Best result for each metric column is in **bold**. '—' indicates data not available.

| | YOLOv8 — ID: BDD | | | | RT-DETR — ID: BDD | | | |
| | COCO-Farther (OOD) | | OI-Farther (OOD) | | COCO-Farther (OOD) | | OI-Farther (OOD) | |
| Method | AUC↑ (%) | FPR95↓ (%) | AUC↑ (%) | FPR95↓ (%) | AUC↑ (%) | FPR95↓ (%) | AUC↑ (%) | FPR95↓ (%) |
|---|---|---|---|---|---|---|---|---|
| ViM | — | — | — | — | 89.5 | 30.7 | 95.2 | 15.2 |
| Mahalanobis | **98.2** | **7.8** | **99.6** | **1.3** | **99.1** | 5.0 | **99.7** | 1.1 |
| MSP | 69.4 | 77.1 | 69.4 | 75.4 | 79.4 | 60.9 | 85.1 | 57.2 |
| Energy | 64.8 | 91.1 | 62.8 | 91.5 | 57.9 | 97.4 | 64.4 | 96.2 |
| ASH | — | — | — | — | 33.1 | 98.6 | 35.4 | 99.2 |
| DICE | — | — | — | — | 60.7 | 90.8 | 58.1 | 96.4 |
| ReAct | — | — | — | — | 56.5 | 96.8 | 63.2 | 95.0 |
| GEN | 63.8 | 71.9 | 66.8 | 68.8 | 77.1 | 67.9 | 83.8 | 63.3 |
| DICE+ReAct | — | — | — | — | 59.3 | 92.7 | 57.0 | 97.3 |
| DDU | — | — | — | — | 99.1 | **3.5** | 99.6 | **0.6** |
| LaRD | — | — | — | — | 98.8 | 5.3 | 99.4 | 1.4 |

Table 18: OOD detection performance for RT-DETR (ID: PascalVOC). LaRD represents best of (Mahalanobis PCA, KNN PCA, GMM PCA). Higher AUC is better (↑), lower FPR95 is better (↓). Best result per metric column is in **bold**.

| | RT-DETR — PascalVOC | | | | | | | |
| | COCO-Near (OOD) | | COCO-Far (OOD) | | OpenImages-Near (OOD) | | OpenImages-Far (OOD) | |
| Method | AUC↑ (%) | FPR95↓ (%) | AUC↑ (%) | FPR95↓ (%) | AUC↑ (%) | FPR95↓ (%) | AUC↑ (%) | FPR95↓ (%) |
|---|---|---|---|---|---|---|---|---|
| ViM | **96.8** | 10.9 | **90.0** | **35.7** | 74.1 | 59.7 | 87.7 | 39.7 |
| Mahalanobis | 96.6 | **10.8** | 87.2 | 42.4 | **91.7** | 32.6 | **92.0** | **29.9** |
| MSP | 94.2 | 21.7 | 84.5 | 58.3 | 62.7 | 79.0 | 76.7 | 67.3 |
| Energy | 68.1 | 97.7 | 70.8 | 92.6 | 50.1 | 96.3 | 62.8 | 96.7 |
| ASH | 64.7 | 86.2 | 57.5 | 92.9 | 46.8 | 96.6 | 49.3 | 94.3 |
| DICE | 63.4 | 89.7 | 70.7 | 83.0 | 81.9 | 73.7 | 81.3 | 78.2 |
| ReAct | 66.3 | 96.0 | 71.5 | 90.8 | 50.9 | 97.5 | 61.9 | 98.1 |
| GEN | 74.9 | 97.7 | 75.6 | 94.7 | 53.2 | 96.0 | 69.9 | 90.1 |
| DICE+ReAct | 68.0 | 90.0 | 70.1 | 84.1 | 81.5 | 75.4 | 79.0 | 83.0 |
| DDU | 96.4 | 11.9 | 86.7 | 45.2 | 91.2 | **32.2** | 91.4 | 31.5 |
| LaRD | 91.8 | 26.6 | 83.3 | 48.8 | 77.8 | 76.2 | 81.2 | 76.0 |

The evaluation using traditional OOD metrics (AUC/FPR95) reveals a significant method-architecture interaction effect on OOD discrimination performance. While certain methods like GEN demonstrate robust OOD separation on specific architectures (e.g., FasterRCNN), their efficacy is not universally transferable. Conversely, density-based methods like Mahalanobis show high sensitivity to the feature space, achieving exceptional discrimination in some contexts (e.g., YOLOv8/RT-DETR on BDD) but underperforming in others. This variability underscores that current OOD scoring functions often exploit specific architectural properties or data distributions rather than embodying a generalizable principle of OOD detection.

Across the presented experiments, traditional OOD detection metrics like AUC and FPR95 generally indicated that distinguishing out-of-distribution objects becomes less challenging as their semantic distance from the in-distribution data increases. This broad trend falsely suggests that greater dissimilarity simplifies the OOD object detection task. However, these metrics, while useful for gauging overall separability, offer limited insight into if these unknown objects are actually found, or the precision of their identification within an object detection framework.

F.2   DETAILED RESULTS ON THE NEWLY INCORPORATED OSOD METRICS

Table 19: OOD detection performance comparison on COCO splits (ID: PascalVOC). Lower nOSE is better (↓), higher $AP_U$/$P_U$/$R_U$ is better (↑). Best result per metric column is in **bold**.

| | YOLOv8 — PascalVOC | | | | | | |
| | COCO-Near (OOD) | | | | COCO-Far (OOD) | | |
| Method | nOSE↓ | $AP_U$↑ | $P_U$↑ | $R_U$↑ | nOSE↓ | $AP_U$↑ | $P_U$↑ | $R_U$↑ |
|---|---|---|---|---|---|---|---|---|
| MSP | 32.3 | 8.5 | 62.0 | 11.0 | 18.7 | 4.5 | **61.1** | 5.7 |
| Energy | 43.6 | 1.3 | 44.3 | 3.0 | 23.2 | 1.0 | 34.8 | 2.7 |
| GEN | 27.2 | 11.7 | 64.7 | 16.5 | 14.2 | 6.3 | 59.3 | 10.1 |
| LaRD | **24.9** | **13.7** | **67.8** | **18.8** | **11.4** | **7.0** | 52.5 | **12.7** |

Table 20: OOD detection performance comparison on OpenImages splits (ID: PascalVOC). Lower nOSE is better (↓), higher $AP_U$/$P_U$/$R_U$ is better (↑). Best result per metric column is in **bold**.

| | YOLOv8 — PascalVOC | | | | | | |
| | OpenImages-Near (OOD) | | | | OpenImages-Far (OOD) | | |
| Method | nOSE↓ | $AP_U$↑ | $P_U$↑ | $R_U$↑ | nOSE↓ | $AP_U$↑ | $P_U$↑ | $R_U$↑ |
|---|---|---|---|---|---|---|---|---|
| MSP | 26.2 | 6.2 | **62.6** | 7.6 | 13.8 | 2.1 | 52.3 | 3.1 |
| Energy | 34.3 | 0.9 | 41.2 | 2.0 | 15.9 | 0.8 | 42.0 | 1.8 |
| GEN | **23.4** | **7.5** | 62.2 | **11.0** | **9.5** | 4.0 | 52.1 | 6.9 |
| LaRD | 26.9 | 5.5 | 60.1 | 8.2 | 9.8 | **4.1** | **52.8** | **7.0** |

Table 21: OOD detection performance comparison on Far OOD sets (ID: BDD). Lower nOSE is better (↓), higher $AP_U$/$P_U$/$R_U$ is better (↑). Best result per metric column is in **bold**.

| | YOLOv8 — BDD | | | | | | |
| | COCO-Farther (OOD) | | | | OpenImages-Farther (OOD) | | |
| Method | nOSE↓ | $AP_U$↑ | $P_U$↑ | $R_U$↑ | nOSE↓ | $AP_U$↑ | $P_U$↑ | $R_U$↑ |
|---|---|---|---|---|---|---|---|---|
| MSP | 4.6 | 0.3 | 31.4 | 1.0 | 4.0 | 0.6 | **36.5** | 1.2 |
| Energy | 5.3 | 0.1 | 26.0 | 0.4 | 5.0 | 0.1 | 22.6 | 0.3 |
| GEN | 3.9 | 0.6 | **34.3** | 1.7 | 3.2 | 0.8 | 36.0 | 2.0 |
| LaRD | **0.1** | **1.6** | 31.1 | **4.8** | **0.0** | **1.4** | 28.3 | **4.7** |

Table 22: OOD detection performance for FasterRCNN (Vanilla) on COCO splits (ID: PascalVOC). Lower nOSE is better ($\downarrow$), higher $AP_U$/$P_U$/$R_U$ is better ($\uparrow$). Best result per metric column is in **bold**. [B]Indicates the primary scoring method of the VOS (Virtual Outlier Synthesis).

| | FasterRCNN (Vanilla) — PascalVOC | | | | | | | |
| | COCO-Near (OOD) | | | | COCO-Far (OOD) | | | |
| Method | nOSE↓ | $AP_U$↑ | $P_U$↑ | $R_U$↑ | nOSE↓ | $AP_U$↑ | $P_U$↑ | $R_U$↑ |
|---|---|---|---|---|---|---|---|---|
| ViM | 38.3 | 5.0 | 69.0 | 6.3 | 17.9 | 1.9 | 56.5 | 2.6 |
| Mahalanobis | 44.6 | 0.2 | 85.7 | 0.2 | 20.6 | 0.1 | **100.0** | 0.1 |
| MSP | 33.5 | 7.4 | 65.4 | 10.2 | 15.2 | 2.8 | 49.5 | 5.2 |
| KNN | 39.6 | 4.2 | 77.0 | 5.0 | 18.9 | 1.0 | 53.2 | 1.7 |
| Energy | 16.0 | 22.3 | 75.9 | 24.8 | 9.8 | 8.0 | 66.3 | 9.9 |
| ASH | 21.0 | 18.1 | 76.4 | 20.5 | 13.5 | 5.6 | 71.2 | 6.6 |
| DICE | 26.7 | 14.2 | 77.4 | 16.2 | 15.3 | 3.9 | 66.2 | 4.9 |
| ReAct | 33.3 | 10.0 | **86.5** | 10.8 | 19.0 | 1.3 | 83.3 | 1.5 |
| GEN | **14.3** | **23.2** | 73.8 | **26.1** | **8.7** | 8.6 | 65.2 | 11.0 |
| DICE+ReAct | 43.0 | 1.3 | 69.6 | 1.8 | 20.2 | 0.3 | 72.7 | 0.4 |
| DDU | 44.3 | 0.3 | 40.5 | 0.5 | 20.2 | 0.3 | 40.0 | 0.4 |
| VOS[B](Energy) | 20.5 | 21.5 | 72.1 | 24.6 | 9.6 | **8.3** | 55.6 | **11.3** |
| LaRD | 39.9 | 3.3 | 65.5 | 4.3 | 17.5 | 2.6 | 71.8 | 3.1 |

Table 23: OOD detection performance for FasterRCNN (Vanilla) on OpenImages splits (ID: PascalVOC). Lower nOSE is better ($\downarrow$), higher $AP_U$/$P_U$/$R_U$ is better ($\uparrow$). Best result per metric column is in **bold**. [B]Indicates the primary scoring method of the VOS (Virtual Outlier Synthesis).

| | FasterRCNN (Vanilla) — PascalVOC | | | | | | | |
| | OpenImages-Near (OOD) | | | | OpenImages-Far (OOD) | | | |
| Method | nOSE↓ | $AP_U$↑ | $P_U$↑ | $R_U$↑ | nOSE↓ | $AP_U$↑ | $P_U$↑ | $R_U$↑ |
|---|---|---|---|---|---|---|---|---|
| ViM | 30.6 | 3.1 | 66.0 | 4.1 | 11.7 | 0.8 | 59.7 | 1.1 |
| Mahalanobis | 35.0 | 0.2 | **100.0** | 0.2 | 12.7 | 0.0 | 0.0 | 0.0 |
| MSP | 28.4 | 4.1 | 59.9 | 6.1 | 9.7 | 1.7 | 53.0 | 2.9 |
| KNN | 33.5 | 1.0 | 57.5 | 1.7 | 11.7 | 0.9 | 62.0 | 1.1 |
| Energy | 18.1 | 12.9 | 73.6 | 15.2 | 5.8 | 5.3 | 70.1 | 6.6 |
| ASH | 21.1 | 10.7 | 75.3 | 12.5 | 7.8 | 3.8 | 69.9 | 4.6 |
| DICE | 23.0 | 9.7 | 75.8 | 11.0 | 9.0 | 2.9 | 70.0 | 3.5 |
| ReAct | 28.4 | 5.7 | 86.4 | 6.1 | 11.8 | 0.8 | **78.7** | 0.9 |
| GEN | **15.9** | **14.1** | 72.0 | **16.9** | **5.4** | 5.4 | 68.0 | 6.9 |
| DICE+ReAct | 33.4 | 1.5 | 82.4 | 1.7 | 12.7 | 0.0 | 50.0 | 0.0 |
| DDU | 34.5 | 0.5 | 51.6 | 0.6 | 12.6 | 0.1 | 46.2 | 0.1 |
| VOS[B](Energy) | 22.3 | 10.3 | 64.1 | 12.8 | 6.3 | **5.5** | 67.3 | **7.1** |
| LaRD | 31.1 | 3.4 | 74.6 | 3.7 | 10.0 | 2.2 | 68.8 | 2.6 |

Table 24: OOD detection performance for FasterRCNN (Vanilla) on Far OOD sets (ID: BDD). Lower nOSE is better ($\downarrow$), higher $AP_U/P_U/R_U$ is better ($\uparrow$). Best result per metric column is in **bold**. [B]Indicates the primary scoring method of the VOS (Virtual Outlier Synthesis).

| | **FasterRCNN (Vanilla) — BDD** | | | | | | |
| | **COCO-Farther (OOD)** | | | | **OpenImages-Farther (OOD)** | | |
| Method | nOSE$\downarrow$ | $AP_U\uparrow$ | $P_U\uparrow$ | $R_U\uparrow$ | nOSE$\downarrow$ | $AP_U\uparrow$ | $P_U\uparrow$ | $R_U\uparrow$ |
|---|---|---|---|---|---|---|---|---|
| ViM | 1.1 | 1.2 | 22.9 | 3.9 | 0.7 | 0.9 | 18.3 | 3.3 |
| Mahalanobis | 2.0 | 1.0 | 21.4 | 3.1 | 2.4 | 0.4 | 11.8 | 1.8 |
| MSP | 3.3 | 0.3 | 17.8 | 1.9 | 2.4 | 0.3 | 14.9 | 1.7 |
| KNN | 1.9 | 1.0 | 23.3 | 3.2 | 0.7 | 1.1 | 20.5 | 3.3 |
| Energy | 2.0 | 0.9 | 22.9 | 3.0 | 0.6 | 1.2 | 22.1 | 3.4 |
| ASH | 3.3 | 0.5 | 20.5 | 1.8 | 2.1 | 0.6 | 19.0 | 2.1 |
| DICE | 2.3 | 0.8 | 22.7 | 2.8 | 1.0 | 1.0 | 21.4 | 3.0 |
| ReAct | 4.0 | 0.4 | 17.9 | 1.2 | 3.6 | 0.1 | 7.7 | 0.6 |
| GEN | 2.0 | 1.0 | 22.9 | 3.0 | 0.7 | 1.2 | 21.8 | 3.3 |
| DICE+ReAct | 4.3 | 0.1 | 14.4 | 0.9 | 3.7 | 0.0 | 7.3 | 0.5 |
| DDU | 3.2 | 0.6 | 19.7 | 2.0 | 3.2 | 0.1 | 9.4 | 1.0 |
| VOS[B](Energy) | 1.8 | **1.8** | **26.7** | **4.7** | 0.6 | **2.2** | **26.2** | **5.6** |
| LaRD | **0.7** | 1.3 | 21.0 | 4.2 | **0.6** | 0.8 | 16.5 | 3.4 |

Table 25: OOD detection performance for FasterRCNN (VOS) on COCO splits (ID: PascalVOC). Lower nOSE is better ($\downarrow$), higher $AP_U/P_U/R_U$ is better ($\uparrow$). Best result per metric column is in **bold**. [B]Indicates the primary scoring method of the VOS (Virtual Outlier Synthesis).

| | **FasterRCNN (VOS) — PascalVOC** | | | | | | |
| | **COCO-Near (OOD)** | | | | **COCO-Far (OOD)** | | |
| Method | nOSE$\downarrow$ | $AP_U\uparrow$ | $P_U\uparrow$ | $R_U\uparrow$ | nOSE$\downarrow$ | $AP_U\uparrow$ | $P_U\uparrow$ | $R_U\uparrow$ |
|---|---|---|---|---|---|---|---|---|
| ViM | 45.5 | 3.1 | 64.0 | 4.3 | 20.3 | 1.3 | 48.9 | 2.2 |
| Mahalanobis | 50.0 | 0.0 | 0.0 | 0.0 | 22.6 | 0.0 | 0.0 | 0.0 |
| MSP | 39.6 | 7.0 | 66.6 | 9.6 | 17.6 | 2.6 | 44.5 | 4.7 |
| KNN | 36.3 | 10.3 | 73.9 | 12.2 | 14.9 | 4.7 | 55.7 | 6.9 |
| ASH | 17.6 | 23.3 | 71.3 | 26.5 | 9.5 | 8.7 | 60.4 | 11.4 |
| DICE | 33.0 | 12.4 | 73.6 | 15.2 | 14.1 | 5.1 | 56.6 | 7.4 |
| ReAct | 42.4 | 6.6 | **83.6** | 6.8 | 20.7 | 1.3 | **66.0** | 1.7 |
| GEN | **15.9** | **24.1** | 69.4 | **27.8** | **8.0** | **9.1** | 54.9 | **12.7** |
| DICE+ReAct | 50.0 | 0.0 | 0.0 | 0.0 | 22.6 | 0.0 | 0.0 | 0.0 |
| DDU | 49.8 | 0.2 | 46.7 | 0.2 | 22.3 | 0.2 | 25.0 | 0.3 |
| VOS[B](Energy) | 20.5 | 21.5 | 72.1 | 24.6 | 9.6 | 8.3 | 55.6 | 11.3 |
| LaRD | 42.1 | 6.0 | 70.7 | 7.1 | 19.9 | 2.1 | 64.5 | 2.5 |

Table 26: OOD detection performance for FasterRCNN (VOS) on OpenImages splits (ID: PascalVOC). Lower nOSE is better ($\downarrow$), higher $AP_U/P_U/R_U$ is better ($\uparrow$). Best result per metric column is in **bold**. [B]Indicates the primary scoring method of the VOS (Virtual Outlier Synthesis).

| | **FasterRCNN (VOS) — PascalVOC** | | | | | | |
| | **OpenImages-Near (OOD)** | | | | **OpenImages-Far (OOD)** | | |
| Method | nOSE$\downarrow$ | $AP_U\uparrow$ | $P_U\uparrow$ | $R_U\uparrow$ | nOSE$\downarrow$ | $AP_U\uparrow$ | $P_U\uparrow$ | $R_U\uparrow$ |
|---|---|---|---|---|---|---|---|---|
| ViM | 34.9 | 1.5 | 49.1 | 2.2 | 13.2 | 0.4 | 46.3 | 0.6 |
| Mahalanobis | 37.3 | 0.0 | 0.0 | 0.0 | 13.8 | 0.0 | 0.0 | 0.0 |
| MSP | 31.7 | 3.0 | 53.2 | 5.4 | 10.9 | 1.4 | 49.1 | 2.7 |
| KNN | 31.6 | 3.7 | 58.7 | 5.3 | 8.9 | 3.5 | 63.5 | 4.6 |
| ASH | 20.7 | 11.9 | 65.5 | 14.1 | 7.1 | 4.8 | 65.5 | 6.3 |
| DICE | 28.9 | 6.0 | 62.9 | 7.6 | 9.4 | 3.1 | 63.3 | 4.2 |
| ReAct | 32.2 | 4.4 | **84.2** | 4.7 | 12.7 | 0.9 | **72.1** | 1.1 |
| GEN | **18.4** | **12.9** | 63.3 | **15.9** | **5.9** | **5.7** | 66.1 | **7.5** |
| DICE+ReAct | 37.3 | 0.0 | 0.0 | 0.0 | 13.8 | 0.0 | 0.0 | 0.0 |
| DDU | 37.3 | 0.0 | 0.0 | 0.0 | 13.7 | 0.0 | 21.4 | 0.1 |
| VOS[B](Energy) | 22.3 | 10.3 | 64.1 | 12.8 | 6.3 | 5.5 | 67.3 | 7.1 |
| LaRD | 32.8 | 3.7 | 75.0 | 4.1 | 11.3 | 1.9 | 73.1 | 2.3 |

Table 27: OOD detection performance for FasterRCNN (VOS) on Far OOD sets (ID: BDD). Lower nOSE is better ($\downarrow$), higher $AP_U/P_U/R_U$ is better ($\uparrow$). Best result per metric column is in **bold**. [B]Indicates the primary scoring method of the VOS (Virtual Outlier Synthesis).

| | **FasterRCNN (VOS) — BDD** | | | | | | |
| | **COCO-Farther (OOD)** | | | | **OpenImages-Farther (OOD)** | | |
| Method | nOSE$\downarrow$ | $AP_U\uparrow$ | $P_U\uparrow$ | $R_U\uparrow$ | nOSE$\downarrow$ | $AP_U\uparrow$ | $P_U\uparrow$ | $R_U\uparrow$ |
|---|---|---|---|---|---|---|---|---|
| ViM | 1.1 | 1.7 | 24.1 | 5.3 | 0.8 | 1.7 | 23.1 | 5.5 |
| Mahalanobis | 2.3 | 1.3 | 22.1 | 4.3 | 3.3 | 0.8 | 17.8 | 3.4 |
| MSP | 4.4 | 0.5 | 19.9 | 2.4 | 3.9 | 0.6 | 21.9 | 2.9 |
| KNN | 2.3 | 1.5 | 24.8 | 4.2 | 0.8 | 2.0 | 26.6 | 5.5 |
| ASH | 3.6 | 1.0 | 23.7 | 3.0 | 2.4 | 1.6 | 26.7 | 4.1 |
| DICE | 2.8 | 1.4 | 25.9 | 3.8 | 1.1 | **2.3** | **29.1** | 5.2 |
| ReAct | 3.6 | 1.1 | 24.0 | 3.1 | 4.7 | 0.5 | 17.6 | 2.1 |
| GEN | 1.6 | 1.8 | 26.3 | 4.8 | 0.6 | 2.1 | 25.5 | 5.6 |
| DICE+ReAct | 5.4 | 0.3 | 16.7 | 1.4 | 5.9 | 0.1 | 13.4 | 1.0 |
| DDU | 3.8 | 0.7 | 20.6 | 2.9 | 4.8 | 0.3 | 14.9 | 2.1 |
| VOS[B](Energy) | 1.8 | 1.8 | **26.7** | 4.7 | 0.6 | 2.2 | 26.2 | 5.6 |
| LaRD | **0.4** | **2.0** | 23.5 | **6.0** | **0.0** | 1.8 | 21.8 | **6.3** |

Table 28: OOD detection performance for RT-DETR on COCO splits (ID: PascalVOC). Lower nOSE is better ($\downarrow$), higher $AP_U/P_U/R_U$ is better ($\uparrow$). Best result per metric column is in **bold**.

| | **RT-DETR — PascalVOC** | | | | | | |
| | **COCO-Near (OOD)** | | | | **COCO-Far (OOD)** | | |
| Method | nOSE$\downarrow$ | $AP_U\uparrow$ | $P_U\uparrow$ | $R_U\uparrow$ | nOSE$\downarrow$ | $AP_U\uparrow$ | $P_U\uparrow$ | $R_U\uparrow$ |
|---|---|---|---|---|---|---|---|---|
| MSP | 2.9 | 20.0 | 96.4 | 20.8 | 2.3 | 4.4 | 87.7 | 5.1 |
| ViM | 4.2 | 18.9 | 96.4 | 19.6 | 1.5 | 5.5 | 92.1 | 5.9 |
| Mahalanobis | 0.9 | 21.3 | 93.6 | 22.8 | 0.7 | 5.5 | 83.9 | 6.6 |
| KNN | **0.8** | **21.4** | 93.0 | **23.0** | **0.4** | **5.6** | 80.8 | **6.8** |
| Energy | 23.8 | 0.0 | 0.0 | 0.0 | 7.5 | 0.0 | 0.0 | 0.0 |
| ASH | 22.2 | 1.5 | 89.5 | 1.6 | 7.4 | 0.0 | 16.7 | 0.1 |
| DICE | 23.7 | 0.1 | **100.0** | 0.1 | 7.2 | 0.3 | **100.0** | 0.3 |
| ReAct | 23.8 | 0.0 | 0.0 | 0.0 | 7.4 | 0.1 | **100.0** | 0.1 |
| GEN | 23.8 | 0.0 | 0.0 | 0.0 | 7.5 | 0.0 | 0.0 | 0.0 |
| DICE+ReAct | 23.6 | 0.2 | **100.0** | 0.2 | 6.9 | 0.5 | 90.9 | 0.5 |
| DDU | 1.2 | 21.2 | 94.1 | 22.5 | 1.0 | 5.5 | 86.3 | 6.4 |
| LaRD | 5.7 | 17.1 | 95.4 | 17.9 | 3.2 | 3.2 | 75.7 | 4.2 |

Table 29: OOD detection performance for RT-DETR on OpenImages splits (ID: PascalVOC). Lower nOSE is better ($\downarrow$), higher $AP_U/P_U/R_U$ is better ($\uparrow$). Best result per metric column is in **bold**.

| | **RT-DETR — PascalVOC** | | | | | | |
| | **OpenImages-Near (OOD)** | | | | **OpenImages-Far (OOD)** | | |
| Method | nOSE$\downarrow$ | $AP_U\uparrow$ | $P_U\uparrow$ | $R_U\uparrow$ | nOSE$\downarrow$ | $AP_U\uparrow$ | $P_U\uparrow$ | $R_U\uparrow$ |
|---|---|---|---|---|---|---|---|---|
| MSP | 24.6 | 4.8 | 69.6 | 6.9 | 10.8 | 3.1 | 58.0 | 5.3 |
| ViM | 23.1 | 6.7 | 77.9 | 8.5 | 8.8 | 5.7 | 72.0 | 7.3 |
| Mahalanobis | **6.0** | **21.8** | 78.5 | **24.8** | **2.9** | **9.6** | 66.3 | **12.6** |
| KNN | 7.8 | 19.7 | 76.7 | 23.0 | 3.3 | 9.0 | 62.9 | 12.2 |
| Energy | 31.7 | 0.0 | 0.0 | 0.0 | 16.4 | 0.0 | 0.0 | 0.0 |
| ASH | 31.6 | 0.0 | 8.7 | 0.1 | 16.1 | 0.1 | 31.6 | 0.3 |
| DICE | 29.5 | 2.1 | **91.5** | 2.2 | 15.1 | 1.1 | **81.0** | 1.3 |
| ReAct | 31.7 | 0.0 | 0.0 | 0.0 | 16.4 | 0.0 | 0.0 | 0.0 |
| GEN | 31.7 | 0.0 | 0.0 | 0.0 | 16.4 | 0.0 | 0.0 | 0.0 |
| DICE+ReAct | 28.5 | 3.1 | 90.9 | 3.2 | 15.1 | 1.0 | 76.9 | 1.2 |
| DDU | 7.3 | 20.3 | 77.2 | 23.5 | 3.9 | 9.0 | 66.2 | 11.7 |
| LaRD | 27.3 | 3.0 | 66.5 | 4.2 | 13.2 | 1.6 | 47.9 | 3.0 |

Table 30: OOD detection performance for RT-DETR on Far OOD sets (ID: BDD). Lower nOSE is better (↓), higher $AP_U$/$P_U$/$R_U$ is better (↑). Best result per metric column is in **bold**.

| | RT-DETR — BDD | | | | | | |
| | COCO-Farther (OOD) | | | OpenImages-Farther (OOD) | | | |
| Method | nOSE↓ | $AP_U$↑ | $P_U$↑ | $R_U$↑ | nOSE↓ | $AP_U$↑ | $P_U$↑ | $R_U$↑ |
|---|---|---|---|---|---|---|---|---|
| MSP | 15.4 | 4.2 | 35.6 | 11.8 | 7.2 | 1.9 | 18.3 | 10.1 |
| ViM | 14.1 | 4.8 | 34.5 | 12.8 | 5.4 | 1.7 | 14.4 | 11.6 |
| Mahalanobis | **0.2** | 11.2 | 33.0 | **25.0** | **0.0** | 2.3 | 12.4 | **14.9** |
| KNN | 0.4 | 11.4 | 33.0 | 24.8 | **0.0** | 2.4 | 12.5 | **14.9** |
| Energy | 28.6 | 0.0 | 0.0 | 0.0 | 20.6 | 0.0 | 0.0 | 0.0 |
| ASH | 28.5 | 0.0 | 19.1 | 0.1 | 20.6 | 0.0 | 10.5 | 0.0 |
| DICE | 27.9 | 0.5 | **77.5** | 0.7 | 20.5 | 0.0 | **36.4** | 0.1 |
| ReAct | 28.6 | 0.0 | 7.1 | 0.0 | 20.6 | 0.0 | 25.0 | 0.0 |
| GEN | 27.9 | 0.4 | 54.2 | 0.7 | 20.0 | 0.2 | 33.9 | 0.4 |
| DICE+ReAct | 27.7 | 0.7 | 70.9 | 0.9 | 20.6 | 0.0 | 25.0 | 0.1 |
| DDU | 0.6 | **11.5** | 34.1 | 24.7 | **0.0** | **2.4** | 12.6 | **14.9** |
| LaRD | 0.8 | 10.7 | 32.3 | 24.4 | **0.0** | 2.3 | 12.4 | **14.9** |

Looking at the results, we don't find a universally best method, neither across architecture nor across semantic distance, e.g: GEN frequently demonstrates strong performance on FasterRCNN (Vanilla and VOS) and YOLOv8 when PascalVOC is the ID, often achieving leading nOSE, $AP_U$, and $R_U$ values. However, its efficacy sharply declines on the RT-DETR architecture with PascalVOC as the ID. Energy, particularly its VOS variant on FasterRCNN and for Far OOD scenarios on BDD, shows competitive results but generally struggles on YOLOv8 and RT-DETR (ID: PascalVOC), characterized by high nOSE and poor recall of unknowns ($R_U$). LaRD's performance is more varied; it excels on YOLOv8 (especially for Far OOD BDD splits) and demonstrates strength on FasterRCNN for BDD Far OOD detection tasks, often leading in nOSE, $AP_U$, and $R_U$. Conversely, its effectiveness is less prominent on FasterRCNN and RT-DETR architectures when trained on PascalVOC. This work also highlights the performance volatility of OOD-OD methods and offers a comprehensive comparative analysis across architectures and semantic similarity.

The introduction of OSOD metrics (nOSE, $AP_U$, $P_U$, $R_U$) provides a much more nuanced understanding of performance related to semantic distance. These metrics reveal that even if general OOD discrimination (AUC/FPR95) seems satisfactory, the actual ability to comprehensively find OOD objects remains unknown. This challenges the intuition that greater dissimilarity inherently makes all aspects of OOD object detection easier.

### F.3 CORRELATIONS AMONG METRICS

Additionally, in Figure 15 it is possible to find the empirical matrix of correlations among all (old and new) metrics. This matrix is calculated from the overall results previously presented. It shows correlations among metrics across all methods, architectures, and OOD datasets. The figure indicates in general significant but moderate correlations between old metrics and new ones, meaning that the AUROC and FPR95 can be indicative of the performance of OOD-OD methods for finding unknown objects. However the correlations don't have a high absolute value (minimum 0.56 an maximum 0.70), which means that new information is added by the new metrics.

Moreover the results indicate that there is no correlation found between old metrics (AUC & FPR95) and $P_U$. This means the $P_U$ is orthogonal to the previous metrics, and therefore the information measured by $P_U$ is invisible to the old metrics. This reinforces the utility of adding OSOD metrics to the benchmark.

## G    FURTHER DISCUSSION ON THE SIMILARITIES AND DIFFERENCES BETWEEN OOD-OD AND OSOD METHODS

Building upon the detailed presentation of how Out-of-Distribution Object Detection (OOD-OD) methods operate in Section 2 and Section D, which draws from previous works (Du et al., 2022b; Wilson et al., 2023; Ammar et al., 2024; Han et al., 2022), we can conclude that the two approaches

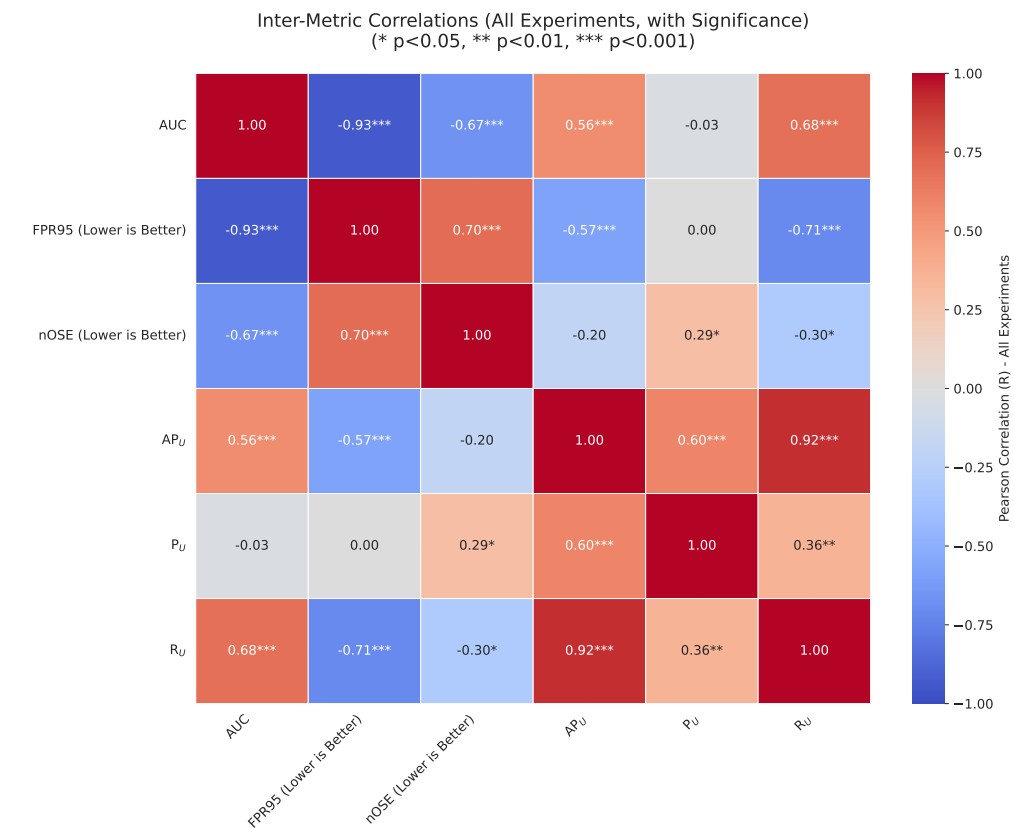

Figure 15: Empirical correlations among old and new metrics.

for handling unknown objects in object detectors are distinct yet they are like two sides of the same coin.

In simpler terms, the current formulation of OOD-OD serves as a monitoring function for the base object detector. It aims to verify that the detected objects are indeed In-Distribution (ID) categories, rather than actively seeking out unknown objects in images. Nevertheless, it *can* identify unknown objects and label them as Out-of-Distribution (OOD). The ability of OOD-OD methods to detect objects was not assessable in the previous benchmark, but it can now be quantified precisely using the new FMIYC benchmark, which employs OSOD metrics calculated with respect to the ground truth labels.

Conversely, open set object detection (OSOD) methods do not rely on monitoring functions. Instead, they incorporate an "unknown" class directly into the object detector, adding specific loss terms and usually training with labeled or pseudo-labeled examples of "unknowns" (Joseph et al., 2021; Dhamija et al., 2020; Gupta et al., 2022). OSOD has developed several metrics, already presented in Section 4.2 and Section C, to measure how well OSOD methods can identify and localize both unknown and known objects simultaneously. The OOD-OD community lacks this type of evaluation, which we believe can significantly enrich the field and is provided by the present benchmark.

We believe the OOD-OD field has substantial potential for future developments, particularly in enhancing a method's ability to localize unknown objects. The main bottleneck is perhaps the filtering of predictions by the confidence threshold in the base model $\mathcal{M}$ because the model is trained to ignore unknown objects. Therefore, finding ways to encourage models to retrieve more predictions that will be post-processed anyway by OOD scoring functions can be an interesting research direction. This could be done perhaps by adjusting the confidence threshold $t^*$ so that a model can retrieve more objects, rather than just maximizing the mAP of the ID test dataset.

Another research direction that may impact the field is the development of OOD-Od or OSOD methods for VLMs, which have broader semantic knowledge and, therefore, may be able to localize several categories of objects beyond a definite set of ID classes. In any case, precise detection of unknown objects must be rigorously evaluated, since this capability is crucial for applications beyond identifying incorrect predictions. Without proper evaluation, OOD-OD methods lack a realistic assessment of their performance for real-world scenarios.

## H  SOCIETAL IMPACT

This work fosters positive societal impacts by enhancing the safety and trustworthiness of object detection systems in safety-critical applications like autonomous driving and medical imaging. By providing a more rigorous benchmark and nuanced metrics for evaluating how well systems detect out-of-distribution objects, it helps prevent overconfidence in deployed models and pushes the field towards developing AI that is more trustworthy and reliable. However, as systems improve in identifying "unknown" or "novel" entities through enhanced evaluations like this, there are several potential downsides to consider. Enhanced capabilities in detecting unspecified "unknowns" could inadvertently enable more pervasive or intrusive surveillance systems, potentially tracking atypical (though not necessarily illicit) activities or objects without clear justification. Furthermore, if the definition of "known" within the training data or benchmark inherently contains biases, such as curation biases, objects or individuals deviating from these biased norms might be disproportionately flagged as"unknown," leading to unfair scrutiny or misclassification for certain groups. There's also a risk that an over-reliance on these improved systems, even with better benchmarking, could lead to a false sense of safety & security, potentially delaying human intervention when truly critical and unanticipated failures occur, or encouraging the deployment of systems in environments where the range of true "unknowns" far exceeds what any benchmark can capture *i.e.*, existence of *unknown-unknowns* in the wild real-word that cannot be foreseeing by any evaluation benchmark.

## I  DATASHEET FOR DATASETS

Upon acceptance, we will provide the dataset datasheet as suggested by Gebru et al. (2021).

