# OpenReview forum: "FindMeIfYouCan: Bringing Open Set Metrics to $\textit{near}$, $\textit{far}$ and $\textit{farther}$ Out-of-Distribution Object Detection"
_ICLR.cc/2026/Conference — Submitted to ICLR 2026_

### Official Review · Reviewer_CM6C · 2025-10-30

**Soundness:** 2
**Presentation:** 2
**Contribution:** 2
**Rating:** 2
**Confidence:** 4

**Summary:**

This paper reveals flaws in the current OOD-OD evaluation protocol, noting that standard metrics fail to evaluate the ability to find unknown objects, at the same time the existing benchmark violates the assumption of non-overlapping in-distribution classes. To address these issues, the authors manually curate and enhance a benchmark with new semantic splits and integrate established open-set object detection metrics that offer deeper insights into how well methods detect unknown objects, when they overlook them, and when they misclassify them. The evaluation shows current metrics do not reflect actual unknown object localization, while observing that semantically similar OOD objects are easier to localize but more likely to be confused with ID classes, whereas far and farther objects show the opposite characteristics.

**Strengths:**

1.The authors' removal of labeled instances with overlapping categories in the benchmark contributes to enhancing the validation accuracy of OOD-OD methods.

2.The authors categorized the OOD datasets into new subsets based on semantic and visual similarity, which contributes to a fine-grained evaluation of OOD-OD method performance.

**Weaknesses:**

1.The authors state that they "properly evaluate the actual identification of unknown objects by integrating complementary metrics from the OSOD community," and that these OSOD metrics can "measure when unknown objects are ignored, when they are detected, and when they are confounded with ID objects." However, since OOD-OD is fundamentally different from OSOD in its objectives, merely aiming to "identify predictions that do not belong to the ID categories". Why do the authors introduce metrics designed for a different task to "capture the complete picture of model performance when encountering unknown objects"? It is suggested for the authors to provide an explanation for this.

2.The authors point out that "a misconception that may be conveyed by these metrics is that a higher AUROC or lower FPR95 means better localization of OOD objects." However, neither of these metrics relates to localization capability. The AUROC serves to evaluate the overall performance of a classifier, while the FPR95 measures the false positive rate for out-of-distribution samples when the true positive rate for in-distribution (ID) samples is 95%. Why do the authors believe these metrics would lead readers to misinterpret them as indicators of "better localization of OOD objects"? It is suggested for the authors to provide an explanation for this.

3.In Table 1, the authors present the percentage of "Ignored objects" in existing OOD-OD benchmarks, but do not provide the corresponding proportion for the proposed FMIYC benchmark. To demonstrate that the new benchmark effectively alleviates the "Ignored objects" issue, it is recommended that the authors supplement this with a comparative analysis.

4.In the EXPERIMENTS AND RESULTS section, the purpose of the experiment on OBJECT DETECTION ARCHITECTURES remains unclear. Furthermore, the reason for incorporating post-hoc methods in the OUT-OF-DISTRIBUTION OBJECT DETECTION METHODS experiment is not sufficiently justified. It is recommended that the authors to explicitly elaborate on the objectives of these two experiments and explain how they collectively demonstrate the effectiveness of the proposed metric/benchmark.

5.In the experimental results presented in Tables 4 and 5, several higher-is-better metrics (such as AUROC, R_U, and P_U) generally demonstrate a decreasing trend under the near-to-farther setup. However, nOSE is a lower-is-better metric. Why does this metric also exhibit a decreasing trend in the near-to-farther configuration? It is suggested for the authors to analyze and explain this observation.

6.In addition to adopting OSOD metrics, the authors have introduced a new evaluation metric. However, neither the formulas for the established OSOD metrics nor the proposed nOSE are provided. It is recommended that the authors include the formal definitions of these metrics to help readers for understanding.

**Questions:**

1.Why do the authors introduce OSOD metrics that differ from the objectives of the OOD-OD task to "capture the complete picture of model performance when encountering unknown objects"?

2.Why do the authors believe that the AUROC and FPR95 metrics would lead readers to misinterpret them as indicating "better localization of OOD objects"?

3.Why do the authors conduct experiments on "mAP across architectures" and post-hoc methods? How do these experiments contribute to validating the proposed metrics and benchmark?

4.Why does the nOSE metric also show a decreasing trend in the near-to-farther experimental setup?

---

> ### Author Response · Authors · 2025-11-23
>
> Dear reviewer CM6C. We appreciate that you acknowledge that the stratification into semantically different levels of similarity contributes to a finer-grained evaluation of OOD-OD.
>
> We address your concerns/weaknesses below:
>
> ---
> ---
>
> 1. We address this concern in the *General Response to Common Concerns* for all reviewers. We elaborate more on this here. As stated in sections 2 and 3, we advocate for a safety-critical point of view, where dealing with unknown objects in object detection can have two possible outcomes: they are either ignored or misclassified. OOD-OD attempts to identify when unknown objects are misclassified. However, measuring when they are ignored is equally relevant from the perspective of use cases such as autonomous driving, as depicted in Figure 2. From the OOD detection community, we strongly believe that taking into account the unknown can allow the OOD detection community to get closer to the promise and ultimate goal of safer and more reliable deployment of DNN-based components.
>
> ---
> 2. As presented in Sections 1 and 3, a possible misinterpretation may occur for non-experts in the OOD-OD field, as well as those coming from OOD detection for image classification. The introduction and presentation of OOD-OD papers, along with their qualitative visualization of performances, may not be explicit enough to state that OOD-OD does not seek to actively find OOD objects. However, this is achieved by detecting incorrect predictions, which may or may not overlap with ground-truth instances. The difference between OOD-OD and OSOD is subtle, and this work aims to highlight those differences and potentially bridge the two fields.
>
> ---
> 3. Table 1 contains the percentage of images without a single prediction in the current benchmark, which does not correspond to the percentage of ignored objects, but is certainly related. We acknowledge that this information is currently missing for the newly proposed benchmark, and we will include it in the main paper if sufficient space becomes available. Otherwise, we will include this in the supplementary material. Here we present the requested table with the number of images without a single prediction for each of the subsets in our benchmark:
>
>
>
>     |                 | ID VOC: Near | ID VOC: Far | ID BDD: Farther |
>     |-----------------|--------------|-------------|-----------------|
>     |                 | OI/COCO      | OI/COCO     | OI/COCO         |
>     | Faster RCNN     | 18.28/20.36  | 49.19/55.01 | 59.88/45.22     |
>     | Faster RCNN VOS | 15.64/19.34  | 44.27/51.49 | 54.28/40.42     |
>
>
>
>    The table shows that the images in the near subset get at least one (incorrect) detection in a higher proportion than for the far or the farther subsets. This is aligned with the findings in Figure 5, which show that objects in the “near” subset get more often confused with ID objects; and the contrary happens for “far’ and “farther” objects.
>
> ---

---

> > ### Author Response · Authors · 2025-12-03
> >
> > We update the previous table with the results from the rest of the architectures tested: Yolov8, RTDETR and OWLv2.
> >
> > Percentage of images without predictions in the newly proposed benchmark:
> > |  | ID VOC: Near | ID VOC: Far | ID BDD: Farther |
> > |---|---|---|---|
> > |  | OI/COCO | OI/COCO | OI/COCO |
> > | Faster RCNN | 18.28/20.36 | 49.19/55.01 | 59.88/45.22 |
> > | Faster RCNN VOS | 15.64/19.34 | 44.27/51.49 | 54.28/40.42 |
> > | Yolov8 | 14.98/18.48 | 30.2/42.32 | 70.15/55.79 |
> > | RT-DETR | 18.06/49.66 | 38.85/81.66 | 14.16/8.06 |
> > | OWLv2 | 77.75/94.12 | 94.57/95.31 | 99.65/98.83 |
> >
> > Expanding on what was explained in the previous comment, the table shows that the object detector models ignore more images when moving to the *farther* split. Interestingly, the VLM model OWLv2 is the one that ignores the most of the images. This indicates that this model mistakes OOD objects less frequently for ID ones. The metrics presented in the results section should be interpreted taking this table into consideration, since AUROC cannot reflect the amount of data used to build it, it needs to be reported.
> >
> > For instance the results from OWL in the farther split indicate an AUROC of about 99%. However this metric is built with only 1% of the images, which corresponds to about 20 images. This results illustrate once more the need to quantify how often OOD-OD methods ignore OOD objects, as is one of the core contributions of our paper.
> >
> > The previous information is indeed invisible from previous benchmark, and gives important information about the performance of the algorithms, that no safety-critical application should ignore.
> >
> > Upon acceptance the table will be included in the main paper if there is available space. Otherwise it will be included in the appendix.
> >
> > The authors

---

> ### Author Response · Authors · 2025-11-23
>
> We address your concerns/weaknesses (**part 2**) below:
>
> ---
> ---
>
> 4. In this work, we expand the scope of the covered architectures in previous papers. So far, for OOD-OD, only Faster-RCNN has been used. Here, we also include the popular YOLOv8, the transformer-based RT-DETR, and the VLM OWLv2. This validates our findings across several architectures. Moreover, the post-hoc methods we introduce in Section 5.2 have shown to achieve state-of-the-art results in OOD benchmarks for image classification [Yang, J. NeurIPS 2022;  Zhang, J DMLR 2024]. All the previous OOD-OD methods [Du et al., ICLR 2022; Du et al., NeurIPS 2022; Wilson et al., ICCV 2023] include a subset of them as baselines. The most natural approach is to use post-hoc methods as baselines and include OOD-OD-specific methods, as is done in the paper.
>
> ---
> 5. The behavior of the open set metrics is of most importance, as you note. In lines 376-417, the situation is analyzed. Indeed, the general trend is that all OSOD metrics decrease their value when moving from “near” to the “farther” subsets. Considering the nOSE, the decrease indicates that “near” objects are more likely to get confused with ID objects, whereas the contrary happens for “farther” objects.
>
> ---
>
> 6. Due to space limitations, the detailed definitions and explanations of each metric, both OOD-OD and OSOD, are presented in Appendix C.
>
> ---
> ---
>
> We hope our explanations answer your concerns. If this is the case, we kindly request that you consider raising your score. Otherwise, we will be happy to clarify any of our statements further.
>
> ---
> ### References
>
> Du, X., Wang, Z., Cai, M., & Li, Y. (2022). VOS: Learning What You Don't Know by Virtual Outlier Synthesis. In International Conference on Learning Representations.
>
> Du, X., Gozum, G., Ming, Y., & Li, Y. (2022). Siren: Shaping representations for detecting out-of-distribution objects. Advances in Neural Information Processing Systems, 35, 20434-20449.
>
> Wilson, S., Fischer, T., Dayoub, F., Miller, D., & Sünderhauf, N. (2023). Safe: Sensitivity-aware features for out-of-distribution object detection. In Proceedings of the IEEE/CVF International Conference on Computer Vision (pp. 23565-23576).
>
> Yang, J., Wang, P., Zou, D., Zhou, Z., Ding, K., Peng, W., ... & Liu, Z. (2022). Openood: Benchmarking generalized out-of-distribution detection. Advances in Neural Information Processing Systems, 35, 32598-32611.
>
> Zhang, J., Yang, J., Wang, P., Wang, H., Lin, Y., Zhang, H., Sun, Y., Du, X., Li, Y., Liu, Z., Chen, Y., & Li, H. (2024). OpenOOD v1.5: Enhanced Benchmark for Out-of-Distribution Detection. Journal of Data-centric Machine Learning Research

---

### Official Review · Reviewer_tFn4 · 2025-10-31

**Soundness:** 3
**Presentation:** 3
**Contribution:** 2
**Rating:** 2
**Confidence:** 5

**Summary:**

The authors discuss shortcomings of existing evaluation metrics for Out-Of-Distribution Object Detection (OOD-OD) benchmarks, and propose alternatives that they land from the close field of Open-Set Object Detection (OSOD).
Additionally, they create a new evaluation benchmark that separated unknown classes into near, far and farther.
Also, they combine OOD methods with object detectors and show that they have no clear advantage over one another.
Finally, they show that conclusions drawn by existing OOD-OD metrics are invalid.

**Strengths:**

* The authors indicate that (class-wise) evaluation of object detection methods, such as evaluated via mAP, is inappropriate.

* The authors introduce post-hoc OOD methods into object detectors to perform open-set object detection.

* The authors provide new evaluation benchmarks for OOD-OD that distinguish between semantically close, medium and far unknown classes.

**Weaknesses:**

1. There are some unintuitive claims made by the authors:

   a) In line 47, the authors claim that "OSOD actively attempts to detect unknown objects". This is completely wrong. For example, the evaluation metric in (Dhamija et al. 2020), which the authors cite for this claim, does not make used of $TP_o$.

   b) In line 131, the authors state that a single confidence threshold $t^*$ is used to perform classification. This is incorrect since mAP actually indicates different thresholds for the different classes -- since they are evaluated individually.

   c) In line 137, the authors discuss two cases: an unknown object is detected as one of the known classes (which corresponds to a false positive $FP_o$) or they are correctly ignored (corresponding to a true negative $TN_o$), which is ignored in all valid OSOR evaluation metrics. Hence, these two cases are two aspects of the same evaluation, and it is not clear why they indicate two different problems OOD-OD and OSOD.

   d) In line 177, the authors state that people might believe that "lower FPR95 means better localization of unknown objects". The reviewer disagrees with this statement since detection/localization of unknown objects is in no way part of this evaluation. The bigger issue here is that we cannot compute a false positive **rate** since the number of detected (unknown or background) objects might differ between detectors.


2. The proposed evaluation metrics are wrong, cumbersome, and do not follow their intuition:

   a) The proposed Unknown Recall and Unknown Precision metrics (line 270) have the same issue as FPR95 and AUROC and are, therefore, meaningless. Researchers in the related field of open-set face recognition have solved this issue by defining false positives per image as absolute numbers [Kasim2024], similarly to the AOSE metric.

   b) While the proposed normalized Open Set Error (nOSE) would be valid as it divides by a constant, its advantage over existing metrics is not obvious. On the other hand, it has the disadvantage that unknown objects need to be labeled -- which might be cumbersome since it is not clear what constitutes an unknown object and what not. For example, there is a street sign in the background of figure 2 -- is this an unknown object or not?

   c) The purpose of the nOSE metric is also not fulfilled. While the authors claim that "nOSE assesses the proportion of unknown objects detected as one of the ID classes", this is not true. If background is detected and classified as ID class (such as in figure 2), this also counts into nOSE, so that the nOSE value can be larger than 1 and, therefore, this is not a proportion.




3. The definition of the new benchmark is unintuitive:

   a) It is not entirely clear why we need two separate sets of images, one containing known (plus some background and unknown) and one containing only unknown objects. The evaluation should concentrate on Ground Truth bounding boxes, not on an image level. Instead of "removing overlaps" (line 223), the authors should have labeled bounding boxes of known objects as such.

   b) It is unclear why and how benchmarks are enriched by adding images from COCO and OpenImages (line 241) -- this process needs to be motivated and explained in more detail.

   c) The definition of near, far and farther is misleading. Since ID datasets differ between far and farther (indicated in line 257), results cannot be compared between these evaluation protocols. However, in the conclusion they do exactly this: they compare far and farther.


4. The evaluation could be improved:

   a) While the authors correctly reject AUROC and FPR95 metrics for OOD-OD, they provide results for those in figure 4. This figure, if at all required, should be moved to the supplemental material.

   b) Similarly, in line 431, the authors define VOS as the best method based on the invalid AUROC metric, which makes no sense.

   c) In the figures, the authors use OpIm as abbreviation, which is introduced nowhere, and is easily confused with Oplm.

   d) The authors used a single threshold (based on 95% TPR) to evaluate their system. While this is valid, it limits the evaluation to a specific operating point. Drawing curves for different operating points would provide more insights into the behavior of the models under different requirements.



5. The Discussion section is merely useless since it just repeats what was already stated in previous sections.

6. The citation style should be adapter throughout the paper, making proper use of \cite, \citep and \citet.

[Kasim2024] Kasim and others: "Watchlist Challenge: 3rd Open-set Face Detection and Identification", IJCB 2024.

**Questions:**

A) The authors discuss shortcomings of the evaluation performed in one related paper (Du et al, 2022b). The reviewer agrees with these shortcomings. However, instead of rejecting the entire research direction of OD-OOD, the authors try to bring evaluation metrics from OSOD into OOD-OD. What is the importance of detecting unknown objects if we cannot classify them? Why do we need OD-OOD, in which scenarios would this be preferred over OSOD, which simply ignores unknown objects?

B) While the authors claim (line 209) that "Accurate localization of ground truth unknown objects is a critical aspect", they do not provide any example where such localization would be required. The authors should discuss such scenarios clearly.

C) Also, the proposed metrics do not fulfill their goals. It would be better to rely on existing metrics such as AOSE, or by dividing AOSE by the number of images in the dataset, to make it comparable across benchmarks. The authors need to defend their choices, or move to appropriate evaluation metrics.

If the authors could clarify the above-mentioned points, and change their paper accordingly, the reviewer would consider increasing their rating.

---

> ### Author Response · Authors · 2025-11-23
>
> Dear reviewer tFn4. We appreciate you taking the time to review our work. We think you may have missed key parts of the paper (especially supplementary material) where your concerns are already addressed.
>
> We address your concerns/weakenesses below:
>
> ---
> ---
>
> ### 1. Unintuitive claims
>
> - **a)** We acknowledge some claims can be counterintuitive since the field is complex; however, we believe they are correct, and we aim to further clarify our presentation:
>
>     We acknowledge that the definition of Open Set may differ across domains and has evolved over the years, particularly in Object Detection. As you correctly point out, in the seminal and early work of [Dhamija et al., WACV 2020], they primarily aim to assess the impact of unknown objects on the performance of Object detectors. Therefore, they do not seek to detect them, and do not define True Positive in open conditions ($TP_O$). However, in more recent works on Open Set Object Detection [Han et al., CVPR 2022; Ammar et al., ECCV 2024], the localization of unknown objects is a standard practice. OSOD's goal is formalized as:
>
>    >*“The goal is to detect all known objects and identify unknown objects so that they will not be misclassified in the known classes.”* [Han et al., CVPR 2022].
>
>    To measure the accuracy of unknown class detection, the metric of average precision for the unknowns ($AP_U$) has been used, which requires defining a True Positive of the unknowns ($TP_U$). As described in Appendix Section C, this corresponds to a detection whose intersection over union (IoU) is greater than or equal to 0.5 and is correctly classified as unknown.
> ---
> - **b)** There are three thresholds that are important not to confuse:
>     - The detection class confidence threshold $t^*$ is set as the operating threshold at inference time of the object detection model
>     - The Intersection over Union (IoU) threshold that is used to calculate the Average precision (AP) metric per class, which, as you said, is calculated through different IoU thresholds. Then, the mean average precision (mAP) is the mean across classes of the AP [Padilla et al, IWSSIP 2020].
>     - The OOD detection function threshold ($\tau$), which is introduced in section 2.2, lines 141 to 144, and explained in detail in Appendix section C.
> ---
> - **c)** For this answer, we provide the following confusion matrix for multiclass classification for object detection. The table is constructed in accordance with best practices from [Tharwat, et al. ACI 2021]. We hope this will clarify the discussion.
>
>
>     |  |  | Predicted |  |  |
>     |:---:|---|:---:|:---:|:---:|
>     |  |  | Unknown | Known | Ignored |
>     |  | Unknown (U) | $TP_U$ | $FN_U^M$ | $FN_U^D$ |
>     | **Ground-truth** | Known (K) | $FP_U$ | $TP_K$ | $FN_K$ |
>     |  | Unlabeled (background) | $FP_U$ | $FP_K$ | - |
>
>     The table focuses on the unknown class, denoted by the U underscore. Ignored objects that are also unlabeled is not possible to be quantified since there are infinite zones in an image that should not be predicted.
>
>     As stated in the previous response 1.a, in the current formulation of OSOD [Liang et al., CVPR 2023; Ammar et al., ECCV 2024], the unknown objects are labeled instances, and one objective is to identify them. In this case, labeled instances of unknowns that are correctly detected as such are the true positives for the unknown class ($TP_U$). This allows for the definition of recall for the unknowns, precision for the unknowns, and the average precision for the unknowns, as presented in Appendix Section C. Furthermore, as presented in Section 2.2, the two areas of OOD-OD and OSOD address the problem of encountering unknown objects for object detectors. Both approaches differ in methodology, evaluation, and metrics, but share a common goal: enhancing the trustworthiness of an object detector when encountering unknown objects. OSOD focuses more on increasing the $TP_U$, while OOD-OD focuses more on reducing the $FN_U^M$ and reducing the $FP_K$. We argue that both approaches are complementary and relevant.
>
> ---
> - **d)** As presented in Section 3, indeed, the localization of unknown objects isn’t the goal of the current evaluation. We challenge this, arguing that the localization of unknown objects is relevant for safety-critical applications. Furthermore, a false positive rate is possible to be computed in the OOD-OD metrics, since the ID and OOD datasets are separated, and OOD-OD is defined as a binary classification task, as presented in section 2.2, and discussed in detail in appendix section D. However, for the OSOD metrics it is indeed not possible to build a false positive rate, due to the fact that for object detection it’s not feasible to quantify the absence of predictions for unlabelled objects [Padilla et al, IWSSIP 2020]. This is another reason for defending the complementarity of the metrics.

---

> ### Author Response · Authors · 2025-11-23
>
> We address your concerns/weakenesses below (**part 2**):
>
> ---
> ---
>
> ### 2. The proposed evaluation metrics are cumbersome
>
> All new metrics have been introduced in section 4.2 and presented in full detail in the appendix section C.
>
> - **a)** It is unclear what issue you are referring to regarding the newly proposed metrics. The main issue we identify with the AUROC and FPR metrics is that they don’t quantify the actual localisation of unknown objects. This issue is remedied with the OSOD metrics of $P_U$, $R_U$, and nOSE, which is one of the main points of our paper. If this is not the issue you are referring to, please clarify. Moreover, upon reviewing the mentioned paper [Kasim2024], we were unable to find any references to metrics using absolute numbers. Can you clarify your statement?
> - **b)** As presented in Section 4.2, the main advantage of the proposed nOSE metric is its comparability across architectures and datasets, which is not possible with the AOSE. Moreover, as you point out, the definition of what constitutes an unknown has been a longstanding debate [Ammar et al., ECCV 2024]. The use of a standard dataset provides a reference: the unknown objects are those labeled instances that do not belong to the ID classes.
> - **c)** As stated in the previous paragraph, the unknown objects in this dataset are defined as the labeled instances of objects not included in the ID classes. The definition of the nOSE is provided in Appendix Section C as: $\frac{FN_U^M}{TP_U + FN_U}$, where $FN_U=FN_U^M +FN_U^D$. The total false negatives for the unknown class are those that are misclassified as one of the ID classes ($FN_U^M$) plus those dismissed ($FN_U^M$). Therefore, within this definition, the nOSE can never be greater than one and constitutes a proportion. The case you cite, where a background proportion is predicted as a known class, corresponds to a false positive for the known classes ($FP_K$), and does not count in the nOSE.
> ---
> ### 3. benchmark definition unintuitive
>
> We provided all possible details to explain our new benchmark both in the main paper and the appendix:
>
> - **a)** The separation between ID and OOD datasets is a well-established procedure in OOD detection [Zhang et al., NeurIPS 2023; Yang et al., IJCV 2023; Yang et al., IJCV 2024]. The ID datasets contain only ID classes, and the OOD datasets contain only OOD classes. Only in this way is it possible to calculate the AUROC and FPR metrics. The evaluation is not concentrated on image-level, but on the object level, as explained throughout the paper, especially in sections 2, 3, and 4. The bounding boxes of the known and unknown objects already exist in the datasets, as they are widely recognized and used for object detection.
>
> - **b)** As explained in Section 4, the current benchmark uses subsets of COCO and OpenImages as OOD datasets. In our work, we not only curate the semantic overlaps and create semantic splits, but we also add more images from the original datasets (not previously present) to increase the number of images in the OOD datasets, thereby enhancing the robustness of the evaluation.
>
> - **c)** We agree that the distinction of semantic similarity could be reframed as near, mid, and far. We propose this semantic stratification following the practice of OOD detection in Image classification [Zhang et al, NeurIPS 2023; Yang et al, IJCV 2024], which already stratifies into near and far. The results are completely comparable, and it is in their comparison that lies one of our main contributions: the fine-grained observations made in sections 5.3 and 6.
> ---
>
> ### 4. evaluation improvement
>
> - As clearly said in Figure 1 and Section 4.2, we insist that we do not reject AUROC and FPR as metrics for OOD-OD. We believe they are completely relevant, and we complement them with OSOD metrics. This is why they are present in the main paper and in the supplementary section in more detail.
> - As stated in the previous paragraph, we never reject AUROC and FPR metrics, so they can be used to compare methods.
> - We acknowledge that the OpIm abbreviation is not correctly introduced in figures 4 and 5, and we will update the figure description to clarify this.
> - The use of the 95% threshold is standard in the OOD detection literature [Du et al., ICLR 2022; Djurisic et al., ICLR 2022; Zhang et al., NeurIPS 2023; Yang et al., IJCV 2024]. Having more thresholds is not the most relevant consideration, as it is not a common practice and may clutter the results or the appendix section.
>
> ---
> ### 5. Discussion section comments
>
> We believe the word “useless” is not very respectful in your comment. Our discussion section summarized our core contributions in light of the presented problems.
>
> ---
> ### 6. Proper citation style
>
> We acknowledge that we can improve the use of the \citet and \citep commands. We will update the version of the paper.

---

> ### Author Response · Authors · 2025-11-23
>
> We address your **questions** below:
>
> ---
> ---
>
> ### Q. A
>
> We insist, as discussed in the previous points, that we do not reject OOD-OD as an area of interest. Our work aims to provide a more sound and holistic evaluation of OOD-OD. Moreover, as is tradition in the OOD and OSOD literature, OOD objects are not to be classified [Ammar et al, ECCV 2024; Han et al, CVPR 2022]. This is the core definition of unknown objects: those that do not belong to the training classes. As explained in Section 2, OOD-OD methods aim to detect incorrect predictions (which may or may not correspond to a labeled unknown object), whereas OSOD methods aim to detect unknown objects themselves (without aiming to detect incorrect predictions). One possible research direction is to integrate these two methods to perform both tasks simultaneously, but this is beyond the scope of our paper.
>
> ---
>
> ### Q. B
>
> We provided the main examples in Figures 1 and 2. In Figure 2, clearly, failing to detect the camels on the road can be seen as a safety-critical situation. Throughout the paper, we argue that detecting incorrect predictions and detecting unknown objects are two sides of the same coin: how object detectors respond when facing unknown objects. The distinction of the failure cases is depicted in the confusion matrix in our answer to concern 1.c of this rebuttal.
>
> ---
>
> ### Q. C
>
> As explained in section 4.2 and answers 2.b and 2.c of this rebuttal, the nOSE is a proposed metric that divides the AOSE by the number of labelled instances of unknown objects. This makes this metric comparable across datasets. Dividing the AOSE by the number of images does not make sense because each image has a different number of labelled instances.
>
> ---
>
> We hope our explanations answer your concerns & questions. If this is the case, we kindly request that you consider raising your score. Otherwise, we will be happy to clarify any of our statements further.
>
> ---
>
> ## References
>
> Du, X., Wang, Z., Cai, M., & Li, Y. (2022). VOS: Learning What You Don't Know by Virtual Outlier Synthesis. In International Conference on Learning Representations.
>
> Han, J., Ren, Y., Ding, J., Pan, X., Yan, K., & Xia, G. S. (2022). Expanding low-density latent regions for open-set object detection. In Proceedings of the IEEE/CVF Conference on Computer Vision and Pattern Recognition (pp. 9591-9600).
>
> Dhamija, A., Gunther, M., Ventura, J., & Boult, T. (2020). The overlooked elephant of object detection: Open set. In Proceedings of the IEEE/CVF winter conference on applications of computer vision (pp. 1021-1030).
>
> Djurisic, A., Bozanic, N., Ashok, A., & Liu, R. (2022). Extremely Simple Activation Shaping for Out-of-Distribution Detection. In The Eleventh International Conference on Learning Representations.
>
> Liang, W., Xue, F., Liu, Y., Zhong, G., & Ming, A. (2023). Unknown sniffer for object detection: Don't turn a blind eye to unknown objects. In Proceedings of the IEEE/CVF conference on computer vision and pattern recognition (pp. 3230-3239).
>
> Padilla, R., Netto, S. L., & Da Silva, E. A. (2020, July). A survey on performance metrics for object-detection algorithms. In 2020 international conference on systems, signals and image processing (IWSSIP) (pp. 237-242). IEEE.
>
> Tharwat, A. (2021). Classification assessment methods. Applied computing and informatics, 17(1), 168-192.
>
> Yang, J., Zhou, K., & Liu, Z. (2023). Full-spectrum out-of-distribution detection. International Journal of Computer Vision, 131(10), 2607-2622.
>
> Yang, J., Zhou, K., Li, Y., & Liu, Z. (2024). Generalized out-of-distribution detection: A survey. International Journal of Computer Vision, 132(12), 5635-5662.
>
> Zhang, J., Yang, J., Wang, P., Wang, H., Lin, Y., Zhang, H., ... & Li, H. OpenOOD v1. 5: Enhanced Benchmark for Out-of-Distribution Detection. In NeurIPS 2023 Workshop on Distribution Shifts: New Frontiers with Foundation Models.

---

### Official Review · Reviewer_rPk8 · 2025-11-01

**Soundness:** 2
**Presentation:** 2
**Contribution:** 2
**Rating:** 4
**Confidence:** 3

**Summary:**

The paper provides a critical examination of current out-of-distribution object detection (OOD-OD) benchmarks. It argues that existing evaluation protocols are flawed because they allow models to ignore unknown objects without penalty. Metrics such as AUROC and FPR do not assess a model’s capability to detect or localize unknown objects. This work introduce FindMeIfYouCan (FMIYC), a  benchmark that eliminates dataset overlaps, categorizes out-of-distribution (OOD) samples into near, far, and farther tiers, and extends OSOD evaluation with PU, RU, APU, and a newly proposed nOSE metric. Experiments cover a diverse set of detectors including Faster R-CNN, YOLOv8, RT-DETR, and OWLv2 as well as standard OOD-OD baselines.

**Strengths:**

The paper presents extensive experiments covering a range of modern object-detection architectures and strong OOD baselines, which strengthens the empirical evaluation.

The proposed stratification of semantic similarity (near vs. far) introduces valuable nuance into OOD assessment and allows for more fine-grained analysis than prior work.

The integration of OSOD metrics helps bridge the gap between OOD detection and open-set object detection frameworks, offering a more unified evaluation protocol.

**Weaknesses:**

The discussion on why AUROC and FPR95 are insufficient for OOD object detection is unclear. It would be beneficial to include a concrete example illustrating how these metrics are computed in the OOD-OD setting, and then explicitly demonstrate why they fail to capture the nuances of this task.

The benchmark is constructed primarily from existing datasets, which raises concerns about originality. The work should better justify why repurposing existing data is sufficient and highlight what novel contributions arise beyond dataset aggregation.

The “near/far/farther” categorization is based on class-name heuristics, which may not reliably reflect semantic similarity.

The manuscript needs more discussion on why OSOD metrics are appropriate for OOD object detection. The conceptual relationship between OSOD and OOD-OD should be explained more thoroughly, and the advantages or limitations of adopting these metrics should be clarified.

**Questions:**

Can FMIYC extend beyond VOC/COCO environments to real-world long-tail or continual-learning settings?

---

> ### Author Response · Authors · 2025-11-23
>
> Dear reviewer rPk8. We appreciate your recognition that our work does extensive experimentation across modern architectures and strong OOD baselines, and the fact that integrating OSOD metrics allows for a more comprehensive evaluation protocol.
>
> We address your concerns/weaknesses below:
>
> ---
> ---
>
> 1. In the main paper, we provide a detailed explanation of how AUROC and FPR are computed in the current benchmark, specifically in section 1, lines 83-89, and then in section 2.2, lines 141-156. We also illustrate their limitations in Section 3, particularly in Figures 2 and 1. More details on the metrics can be found in Appendix Section C. For convenience, we emphasize that AUROC and FPR can measure if a given OOD method can accurately classify incorrect predictions (when an OOD object is confused with an ID one). However, they can not measure if OOD objects are correctly localized, or if they are ignored. The situation is depicted in Figure 2, where the object detector misidentifies a car and ignores the camels. The car detection does not correspond to any ground-truth object, and the ignored camels are outside the AUROC and FPR metrics. To alleviate this situation, we propose complementing the AUROC and FPR with the OSOD metrics, which enable the measurement of the localization of OOD objects when they are ignored and when they are correctly localized but confused with ID objects.
>
> ---
>
> 2. Answered in the *General Response to Common Concerns*.
>
> ---
>
> 3. As presented in Section 4.1, we validated the splits based on class names using WordNet and the Wu-Palmer similarity, with the results shown in Appendix Section B. Moreover, as shown in Figure 3, the splits were further validated in the image space using CLIP image embeddings, which have been demonstrated to capture both semantic and stylistic attributes [Mayilvahanan et al, ICLR 2024].
>
> ---
> 4. This concern is addressed in the *General Response to Common Concerns*.
>
> ---
> ### References
>
> Mayilvahanan, P., Wiedemer, T., Rusak, E., Bethge, M., & Brendel, W. (2024). Does CLIP’s generalization performance mainly stem from high train-test similarity?. In The Twelfth International Conference on Learning Representations.

---

> ### Author Response · Authors · 2025-11-23
>
> ### Answer to your question
>
> > *Can FMIYC extend beyond VOC/COCO environments to real-world long-tail or continual-learning settings?*
>
> Yes, FMIYC can extend to real-world long-tail and continual learning settings:
>
> ---
> **1. Long-Tail Settings:**
>
> Our near, far, and farther taxonomy naturally maps to long-tail scenarios. Recent work in long-tailed discovery (ImbaGCD [Li et al], Long-tailed NCD [Zhang et al. TMLR 2023]) has shown that class imbalance presents significant challenges for open-world systems. In long-tail object detection:
>
> - Near-OOD samples represent common objects from the head of the distribution that models frequently encounter.
> - Far-OOD samples correspond to objects from the mid-tail that are semantically related to known classes but are less frequent.
> - Farther-OOD samples represent rare tail classes that are both semantically distant and statistically rare.
>
> Our benchmark's emphasis on diverse semantic shifts directly addresses the challenge that long-tail methods face: distinguishing between rare in-distribution objects (which should be detected despite low frequency) and truly novel objects (which should be rejected). By providing explicit labels for different OOD granularities, FMIYC enables evaluation of whether methods conflate semantic novelty with statistical rarity—a critical failure mode in long-tail scenarios.
>
> ---
>
> **2. Continual Learning Settings:**
>
> FMIYC provides a strong foundation for continual OOD object detection. Recent work on continual category discovery ( IGCD [Zhao et al. ICCV 2023], Happy [Ma et al. NeurIPS 2024]) has established the paradigm of systems that must:
>
> - Detect unknown objects at task t
> - Learn to recognize them at task t+1
> - Continue detecting new unknowns at task t+2
>
> Our benchmark directly supports this progression:
>
> - Task t: A model trained on VOC must detect COCO objects as OOD. Using our annotations, we can evaluate whether it correctly identifies near (semantically similar) vs. farther (semantically distant) unknowns.
> - Task t+1: After incorporating some COCO classes, the model is evaluated on the remaining COCO classes plus new datasets. Our farther-OOD samples (OpenImages, Objects365) provide the next wave of unknowns.
> - Task t+2: The hierarchical structure of our benchmark (VOC → COCO → OpenImages → Objects365) naturally creates a continual learning curriculum with increasing semantic diversity.
>
> Crucially, our explicit semantic shift labels enable researchers to track whether continual learning methods:
>
> - Maintain OOD detection performance on previously encountered semantic shifts
> - Generalize to new types of semantic shifts
> - Suffer from "catastrophic forgetting" in their OOD detection capabilities
>
> ---
>
> ### References
>
> Li, Ziyun, Ben Dai, Furkan Simsek, Christoph Meinel, and Haojin Yang. "Imbagcd: Imbalanced generalized category discovery." arXiv preprint arXiv:2401.05353 (2023). CVPR 2023 Computer Vision in the Wild Workshop.
>
> Chuyu Zhang, Ruijie Xu, and Xuming He. Novel Class Discovery for Long-Tailed Recognition. Transactions on Machine Learning Research, 2023.
>
> Shijie Ma, Fei Zhu, Zhun Zhong, Wenzhuo Liu, Xu-Yao Zhang, and Cheng-Lin Liu. Happy: A Debiased Learning Framework for Continual Generalized Category Discovery. In Advances in Neural Information Processing Systems, volume 37, pages 50850–50875, 2024.
>
> Bingchen Zhao and Oisin Mac Aodha. Incremental Generalized Category Discovery. In Proceedings of the IEEE/CVF International Conference on Computer Vision (ICCV), pages 19137–19147, October 2023.

---

> > ### Comment · Reviewer_rPk8 · 2025-11-28
> >
> > Thank you for the detailed response. Could you clarify whether Figure 9(b) and Figure 3 present the same content?

---

> > > ### Author Response · Authors · 2025-11-28
> > >
> > > Dear reviewer. Thank you for your response. We are glad if you found our answers satisfactory.
> > >
> > > Indeed, to facilitate comparison between the current benchmark and our newly proposed one, regarding the CLIP embeddings analysis, Figure 9(b) in the appendix includes what is already presented in Figure 3.
> > >
> > > Section B of the appendix provides finer details about the construction of the proposed benchmark. One important aspect of its construction is the validation in both the semantic space and the image space of the stratification into near, far, and farther. For the validation in the image space, we used CLIP image embeddings, which have been shown to contain semantic and stylistic attributes [Mayilvahanan et al, NeuRIPS 2023].
> > >
> > > It is evident from Figure 9(a) that the previous benchmark already reveals a difference in semantic similarity between ID and OOD datasets, which is one of the reasons we utilize this existing difference to create the new stratifications.
> > >
> > > ---
> > > We hope our explanations answer your concerns and questions. If this is the case, we kindly request that you consider raising your score. Otherwise, we will be happy to clarify any of our statements further.
> > >
> > > The authors
> > >
> > >
> > > # References
> > >
> > > Mayilvahanan, P., Wiedemer, T., Rusak, E., Bethge, M., & Brendel, W. Does CLIP’s generalization performance mainly stem from high train-test similarity?. In NeurIPS 2023 Workshop on Distribution Shifts: New Frontiers with Foundation Models.

---

### Official Review · Reviewer_XELp · 2025-11-02

**Soundness:** 2
**Presentation:** 2
**Contribution:** 2
**Rating:** 4
**Confidence:** 3

**Summary:**

Here is a full review based on the revised submission.SummaryThis paper introduces FindMeIfYouCan (FMIYC), a new benchmark for Out-of-Distribution Object Detection (OOD-OD). The work identifies and corrects critical flaws in the existing evaluation protocol, namely the failure to penalize ignored unknown objects and the violation of the non-overlapping-class assumption. The FMIYC benchmark is built by meticulously curating new evaluation splits ("near," "far," and "farther") from established datasets (Pascal-VOC, BDD100k, COCO, OpenImages). It integrates metrics from the Open-Set Object Detection (OSOD) community ($AP_U$, $P_U$, $R_U$) and proposes a new metric, normalized Open Set Error (nOSE), to provide a more holistic evaluation. This revised version significantly expands the analysis to include modern architectures like YOLOv8, RT-DETR, and the Vision-Language Model (VLM) OWLv2, providing novel insights into their OOD performance.

**Strengths:**

1. Addresses Fundamental Flaws: The paper clearly identifies and systematically corrects fundamental, long-standing issues in the standard OOD-OD evaluation. By removing semantic overlaps and using metrics that account for ignored objects, it provides a much more reliable and trustworthy evaluation framework.

2. Nuanced "Near/Far" Splits: The stratification of OOD data into "near," "far," and "farther" splits is a key contribution. It enables a more fine-grained analysis, leading to important insights (e.g., semantically near objects are easier to localize but more often confused with ID classes).

3. Adoption of OSOD Metrics: Integrating OSOD metrics ($AP_U$, $P_U$, $R_U$) and proposing nOSE is a major step forward. These metrics provide a much deeper understanding of model behavior (localization, misclassification, and omission) than traditional AUROC/FPR95, which this paper shows can be misleading.

**Weaknesses:**

1. Dependency on Ground Truth Labels: The new metrics, while superior, introduce a significant practical dependency on exhaustive and correct ground truth (GT) labels for all unknown objects. This trade-off is a key limitation, as acquiring such annotations is a major bottleneck and contrary to the spirit of truly open-set evaluation, where unknowns are, by definition, unlabeled.


2. Incremental Contribution: While the VLM analysis adds substantial value, the benchmark's core contribution is a refinement of existing work. It involves curating subsets from existing datasets and adopting metrics from the related OSOD field, rather than generating entirely new data or a fundamentally new evaluation paradigm.

**Questions:**

Please refer to the above weaknesses

---

> ### Author Response · Authors · 2025-11-23
>
> Dear reviewer, XELp. We appreciate your recognition that our work addresses fundamental flaws in the current benchmark and enables a fine-grained evaluation due to semantic stratification.
>
> We address your concerns/weaknesses below:
>
> ---
> ---
>
> ### 1. Dependency on Ground Truth Labels
>
> We proposed using COCO and OpenImages as OOD datasets for several reasons, one of which is to alleviate the questionability of the labels. Subsets of both datasets are the ones already present in the current OOD-OD benchmark. Furthermore, they are widely used by the object detection community, since they are open-source and of high quality. Indeed, OpenImages constitutes one of the most comprehensive object detection datasets, covering approximately 600 categories [Kuznetsova et al., IJCV 2020].
>
> ---
> ### 2. Incremental contribution
>
> We address this concern in the *General Response to Common Concerns*.
>
>
> ---
>
> We hope our explanations answer your concerns. If this is the case, we kindly request that you consider raising your score. Otherwise, we will be happy to clarify any of our statements further.
>
> ---
>
> ## References
>
> Kuznetsova, A., Rom, H., Alldrin, N., Uijlings, J., Krasin, I., Pont-Tuset, J., ... & Ferrari, V. (2020). The open images dataset v4: Unified image classification, object detection, and visual relationship detection at scale. International journal of computer vision, 128(7), 1956-1981.

---

### Author Response · Authors · 2025-11-23
**General Response to Common Concerns - Part 1**

Dear reviewers, we greatly appreciate your comments, which will help us improve our work. We are pleased that you found our work “Addresses Fundamental Flaws” in the current benchmark, as noted by reviewer **CM6C** and **XELp**. The stratification into near, far, and farther was appreciated as a contribution that “enables a more fine-grained analysis,” as noted by reviewers **XELp**, **rPk8**, and **CM6C**.

We have compiled the common concerns you have raised and expect to address them.

## 1) Incremental work / Originality: Reviewers **XELp**, **rPk8**

As you note, the previous benchmark is flawed and obscures key insights, as stated in Section 1 (Introduction) and Section 3 (Pitfalls of the Current OOD-OD Benchmark). Our work serves as a bridge between the OOD-OD and OSOD communities, which share a similar objective. We can summarize our key contributions as follows:

- We identified fundamental flaws in the previous benchmark, specifically its failure to account for unknown objects and semantic overlaps.
- We manually curated for semantic overlaps, using high-quality, extensively used open-source datasets (BDD100k, Pascal VOC, COCO, OpenImages)
- We demonstrate that semantic stratification into near, far, and farther splits enables targeted OOD-OD evaluation and has the potential to facilitate specific OOD-OD or OSOD methods tailored to meet the needs of semantic similarity.
- We introduced OSOD metrics into the benchmark that accurately measure behavior against unknown objects, specifically when methods ignore them and when they are confounded with ID classes.
- We introduced a new metric, the normalized open set error (nOSE), which enables comparison across methods and datasets.
- We adapted OOD detection methods from image classification to object detection architectures.
- We incorporate architectures that were previously ignored in OOD-OD benchmarks, including YOLOv8, RT-DETR, and the OwLv2 VLM architecture.
- We release open-source code for benchmark construction and evaluation, to facilitate adoption and verification.

As you stated previously in your review, we believe that the unique combination of contributions in this work makes it impactful, even if the data is not an entirely new dataset. The manual curation of overlaps and further stratification into semantically meaningful splits utilizes existing high-quality and widely used open-source datasets, which we believe facilitates use, adoption, and trust in the benchmark. Furthermore, the results of current methods show that the proposed evaluation benchmark is challenging, especially with respect to the newly introduced OSOD metrics. This is indicative of the relevance of combining both sets of metrics into a unified benchmark.

The introduction of normalized Open Set Error (nOSE) is one of the contributions of our work. As stated in section 4.2, in contrast to the Absolute Open Set Error (AOSE), the nOSE is interpretable and comparable across methods and datasets. We believe the introduction of this metric can be relevant to the OSOD community, as the AOSE is widely reported [Han et al. CVPR 2022; Liang et al., 2024], yet it remains non-interpretable and non-comparable.


We address your specific concerns below each of your reviews.

## References:

Han, J., Ren, Y., Ding, J., Pan, X., Yan, K., & Xia, G. S. (2022). Expanding low-density latent regions for open-set object detection. In Proceedings of the IEEE/CVF Conference on Computer Vision and Pattern Recognition (pp. 9591-9600).

Liang, S., Wang, W., Chen, R., Liu, A., Wu, B., Chang, E. C., ... & Tao, D. (2024). Object Detectors in the Open Environment: Challenges, Solutions, and Outlook. CoRR.

---

> ### Author Response · Authors · 2025-11-23
> **General Response to Common Concerns - Part 2**
>
> We have compiled the common concerns you have raised and expect to address them.
>
> ## Why OOD-OD metrics are insufficient and OSOD metrics are appropriate for OOD-OD. Reviewers **rPk8**, **CM6C**:
>
> The importance of OOD for object detection in safety-critical applications has been presented in several forms, one of which is given by Wilson et al, ICCV 2023: *“when deployed into the real world, out-of-distribution (OOD) samples that do not belong to the training distribution are likely encountered. Upon encountering OOD samples, DNNs tend to fail silently and produce overconfident erroneous predictions”*. Meanwhile, Du et al., NeurIPS 2022 write: *“failing to detect OOD objects on the road can directly lead to disastrous accidents.”* Moreover, the definition of the OOD-OD goal has been presented as: *“the task of unsupervised out-of-distribution object detection is recently proposed, whose goal is to accurately detect the objects never-seen-before during training”* [Wu & Deng, IEEE TPAMI 2023].
>
> This highlights the discrepancy and lack of clarity regarding what is crucial for safety-critical applications that utilize object detection models deployed in the real world. These definitions inherit from the definition of OOD detection for Image classification, where each image has only one relevant object, and the model's output is exactly one vector of probabilities for each ID class.
>
> Conversely, for object detection, the model is not constrained to give an exact number of predictions. This causes a mismatch in metrics, where AUROC and FPR are binary classification metrics, not localization metrics. An object detection model might overlook unknown objects, in which case a safety-critical situation is ignored. Even if an unknown object is correctly localized and classified, AUROC and FPR do not evaluate this, as depicted in Figure 2 from our paper. AUROC and FPR only evaluate the identification of incorrect predictions, which might or might not correspond to actual unknown objects.
>
> This is the reason why we advocate for the integration of OSOD metrics, which are localisation ones. They actually measure whether unknown objects are localized, ignored, or confused. The results presented in Section 5 demonstrate that the complementary metrics facilitate a holistic understanding of how the object detector and the OOD-OD methods behave with respect to the labeled unknown objects from the semantically stratified OOD datasets.
>
> ## References:
>
> Wilson, S., Fischer, T., Dayoub, F., Miller, D., & Sünderhauf, N. (2023). Safe: Sensitivity-aware features for out-of-distribution object detection. In Proceedings of the IEEE/CVF International Conference on Computer Vision (pp. 23565-23576).
>
> Wu, A., & Deng, C. (2023). TIB: Detecting unknown objects via two-stream information bottleneck. IEEE Transactions on Pattern Analysis and Machine Intelligence, 46(1), 611-625.
>
> Du, X., Gozum, G., Ming, Y., & Li, Y. (2022). Siren: Shaping representations for detecting out-of-distribution objects. Advances in Neural Information Processing Systems, 35, 20434-20449.

---

### Author Response · Authors · 2025-12-03
**Paper Changelog**

We updated the paper following recommendations from reviewers:

1. We improved the use of \citet and \citep commands as requested by reviewer tFn4. We improved the appropriate citations throughout the main paper and the supplementary material.

2. We properly introduce the OpIm abbreviation in Figures 4 and 5, as requested by reviewer tFn4.

3. We added a table summarizing the amount of images in the newly proposed benchmark, as requested by reviewer CM6C. The table is added in Appendix section F.

4. We incorporated the appendix in the same document as the paper, in order to facilitate reading. This section were before in the attached supplementary material file.

5. The supplementary material file now contains only the code.

The authors

---

### Author Response · Authors · 2025-12-03
**Rebuttal summary**

Dear ACs, SACs, PCs

First of all, we deeply regret the incident about the data leak, which indeed impacts the community. Furthermore, it affected the discussion phase for our work. The most concerned reviewers did not comment before the incident, such as reviewers tFn4 and CM6C.

The reviewers agreed that our work “Addresses Fundamental Flaws” in the current benchmark. The stratification into *near, far, and farther* was appreciated as a contribution that enables a more fine-grained analysis. Furthermore, we cover several architectures and methods, which makes our work an extensive benchmark on strong baselines. Moreover, the integration of OSOD metrics helps bridge the gap between OOD detection and open-set object detection frameworks. Finally, the work will be open-source, which facilitates verification and adoption by the community.

Common concerns raised were:
1. Incremental work/Originality
2. Why OOD-OD metrics are insufficient

Other main concerns:

3. Dependency on Ground Truth Labels
4. The “near/far/farther” categorization is based on class-name heuristics, which may not reliably reflect semantic similarity.

Our responses:
1. Our paper addresses a critical gap in the field: the lack of clarity and consensus on what truly matters for safety-critical applications using object detection models in the real world. To resolve these issues, we make several key contributions: we manually curate fundamental flaws in the existing (de facto) benchmark, adopt the established OOD detection framework by stratifying data into near, far, and farther semantic categories, integrate metrics from the related field of Open Set Object Detection (OSOD), evaluate novel architectures previously unexplored in this context, and adapt robust baselines from OOD detection in image classification. Together, these efforts provide a timely and necessary foundation for advancing and achieving the promise of the OOD-OD field for safe and reliable object detector deployment in the real world.
2. As depicted in section 3 of the paper, the current benchmark suffers from fundamental shortcomings, such as semantic overlaps between ID and OOD datasets. Moreover, its metrics alone obscure a safety-critical concern: ignored objects are not taken into account, which can result in catastrophic consequences. The current metrics of AUROC and FPR do not relate to unknown object localization, and do not use ground-truth (GT) labels. To alleviate these issues, we manually curate and enrich the benchmark by incorporating OSOD metrics that enable the quantification of unknown object localization, as well as when objects are ignored or confused with ID classes.

3. We clarified that the dependence of OSOD metrics on GT labels is balanced by the traditional metrics that do not depend on labels.

4. As presented in Section 4.1, we validated the splits based on class names using WordNet and the Wu-Palmer similarity, with the results shown in Appendix Section B. Moreover, as shown in Figure 3, the splits were further validated in the image space using CLIP image embeddings, which have been demonstrated to capture both semantic and stylistic attributes

Further responses to the reviewer's concerns are presented in the remainder of the rebuttal.

We believe special attention should be given to the review from reviewer tFn4. We believe this reviewer missed key parts of the paper, as most of his concerns are already addressed in the paper or its appendix.

The authors

---

### Meta-Review · Area_Chair_G6uh · 2025-12-10

**Summary:**

This paper considers an interesting problem of establishing a new benchmark for OOD detection in object detection task.
However, the reviewers have raised several concerns over this paper:

1. Originality increment: Originality/Incremental Contribution Benchmark relies on existing datasets; core ideas (OSOD metrics, semantic splits) are not entirely novel.

2. Metric Justification: Why are OSOD metrics appropriate for OOD-OD? Why are AUROC/FPR insufficient?

3. Dependency on Ground Truth (GT) Labels OSOD metrics require GT for unknown objects, conflicting with open-set evaluation (unknowns are unlabeled in practice).

4.Details of experiments.

**Reviewer Concerns:**

Concerns addressed:

1.Technical Revisions:

Fixed citation formatting (\citet/\citep) and OpIm abbreviation in figures.

Merged appendix into the main paper for accessibility; supplementary material now only contains code.

Added benchmark image count table and "images without predictions" table (for ignored object comparison).

Validated near/far/farther splits with WordNet/CLIP embeddings .

2.Clarifications Accepted by Reviewers:

Justified OSOD metric integration (complementarity with AUROC/FPR, safety-critical use cases).

Explained benchmark extensibility to long-tail/continual learning (near→far→farther maps to head→mid→tail classes; hierarchical dataset structure supports continual evaluation).

Clarified nOSE’s validity and advantage over AOSE (cross-dataset comparability, interpretability).

Resolved confusion about AUROC/FPR limitations.

Concerns remained:

1.The core concerns about the increment is not addressed. The reviewers do not agree that the authors have proposed a clearly defined new problem setting.

2.This paper does not provide insights into how to solve this problem.

**Reviewer Scores:**

4,4,2,2, no reviewers state the increase of scores.

---

### Decision · Program_Chairs · 2026-01-26

Reject